



# Towards High Resolution Climate Reconstruction Using an Off-line Data Assimilation and COSMO-CLM 5.00 Model

Bijan Fallah[1], Walter Acevedo[2], Emmanuele Russo[1], Nico Becker[1], and Ulrich Cubasch[1]

[1]Institute of Meteorology, Freie Universität Berlin, Carl-Heinrich-Becker Weg 6-10, 12165 Berlin, Germany
[2]Institute of Mathematics, Universität Potsdam, Karl-Liebknecht-Str. 24/25, D-14476 Potsdam, Germany

*Correspondence to:* Bijan Fallah (bijan.fallah@met.fu-berlin.de)

**Abstract.** Paleo-proxy observations have been recently used to constrain the climate models through data assimilation (DA). However, both DA and climate models are computationally very expensive. Moreover, in paleo-DA, the assimilation period is usually too long for a dynamical model to follow the previous analysis state and the chaotic behavior of the model becomes dominant. The majority of the recent paleoclimate studies using DA have performed low or intermediate resolution global

simulations along with an "off-line" DA approach. In an "off-line" DA, the re-initialisation cycle is completely removed after the assimilation step. In this paper, we design a computationally affordable DA to assimilate yearly pseudo and real observations into an ensemble of COSMO-CLM high resolution regional climate model (RCM) simulations over Europe, where the ensemble members slightly differ in boundary and initial conditions. Within a perfect model experiment, the performance of the applied DA scheme is evaluated with respect to its sensitivity to the noise levels of pseudo-observations. It was observed

that the injected bias in the pseudo-observations does linearly impact the DA skill. Such experiments can serve as a tool for selection of proxy records, which can potentially reduce the background error when they are assimilated in the model. Additionally, the sensibility of the COSMO-CLM to the boundary conditions is addressed. The geographical regions, where the model exhibits high internal variability are identified. Two sets of experiments are conducted by averaging the observations over summer and winter. The dependency of the DA skill to different seasons is investigated. Furthermore, the effect of the

spurious correlations within the observation space is studied and the optimal correlation length, within which the observations are assumed to be correlated, is detected. Finally, the real yearly-averaged observations are assimilated into the RCM and the performance is evaluated against a gridded observation dataset. We conclude that the DA approach is a promising tool for creating high resolution yearly analysis quantities. The affordable DA method can be applied to efficiently improve the climate field reconstruction efforts by combining high resolution paleo-climate simulations and the available proxy observations.



# 1 Introduction

It is now well known that many of the long-term processes (millennial scale) in the climate system have a large impact on predicting the climate even for the shorter time scales (decadal, yearly scales) (Evans et al., 2013; Acevedo et al., 2015; Latif et al., 2016; Steiger and Smerdon, 2017). The improvement of future predictions of the climate models, depends largely on

the understanding of such processes. However, cutting-edge science appears to be immature in describing processes, which take place on time scales beyond the yearly cycle of the climate system (Latif et al., 2016). One of the main challenges is the lack of information for longer time-periods, with observational data sets usually covering less than the recent century. Alternative information of past climate behavior whose time scales expand beyond the available observational data sets are indeed required. Two methods are commonly employed for reconstructing the climate prior to the instrumental records: paleo-

proxy reconstructions and climate models (Hakim et al., 2013).

Climate proxy archives (tree ring, coral, sediment and glacial) are examples of indirect climate observations, which suffer from several ill structural conditions (Acevedo et al., 2015; Jones and Mann, 2004). The data recording processes involved in such archives are very complex, encompassing physical, biological and chemical processes (Evans et al., 2013). The temporal resolution of climate proxies does not exceed seasonal time scales. Differentiating the climate and human impact on proxies is

also a challenging work. Furthermore, inverting proxy records into climate information is traditionally done in the framework of statistical modeling and multivariate linear regression techniques dominate this area (Acevedo et al., 2015). Using more sophisticated statistical models is problematic, because the overlapping time span between the instrumental (weather station observations) and the proxy records becomes too short to train the statistical models.

Climate models may serve as an alternative method for the investigation of the long-term paleoclimate variability. They

create dynamically consistent state of the climate system by using numerical methods (Goosse, 2016). However, their reconstructed states are very sensible to the initial conditions and to the imposed forcings, as well as to the parametrisation schemes used for the representation of sub-grid scale processes (small-scale processes which are not explicitly resolved by the model). An improvement of the spatial resolution of climate models is thought to be of crucial importance for the study of past climate changes, in particular when comparing their results against proxy data which are highly affected by local-scale processes

(Renssen et al., 2001; Bonfils et al., 2004; Masson et al., 1999; Fallah and Cubasch, 2015; Russo and Cubasch, 2016; Russo, 2016). As a consequence, along with GCMs, high resolution regional climate models (RCMs) are recently being applied in paleoclimate studies. Several recent studies have applied a time-slice climate simulation method (Kaspar and Cubasch, 2008) to downscale dynamically the global paleo-climate simulations with a higher resolution (Prömmel et al., 2013; Fallah et al., 2016b; Russo and Cubasch, 2016). Simulation of the climate of the past using RCMs is a challenging approach due to their

high computational costs and dependency of such models on the driving general circulation model (GCM). Given the computational costs of RCM, previous studies, which use the time-slice simulation method, could conduct only a single climate run. A single model simulation may not provide a sophisticated measure of the uncertainty.

In addition to the two above-mentioned methodologies, a novel and appealing technique for the reconstruction of the climate of the past is Data Assimilation (DA). According to Talagrand (1997), assimilation of the observations is the process through




which the state of the atmospheric or oceanic flow is estimated by using the available observations and the physical laws which govern its evolution, presented in the form of a numerical model. DA has recently emerged as a powerful tool for paleoclimate studies, mathematically blending together the information from proxy records and climate models (Evensen, 2003; Hughes et al., 2010; Brönnimann, 2011; Bhend et al., 2012; Dee et al., 2016; Hakim et al., 2013; Steiger et al., 2014; Matsikaris et al., 2015, 2016; Hakim et al., 2016; Acevedo et al., 2017; Okazaki and Yoshimura, 2017; Perkins and Hakim, 2017). For a review of the DA techniques which are applied in paleoclimate studies, we refer to the works of Acevedo et al. (2015, 2017) and Steiger and Hakim (2015). One limiting aspect of classical DA methods is the fact that the realistic climate models may not have predictive skills longer than several months (Acevedo et al., 2017; Perkins and Hakim, 2017). The models lose their forecasting skills shortly after initialization and evolve freely until the next step where the observations are available. Considering the complexity of implementation of DA methods into climate models, their high computational expenses and the short forecasting horizon of the models, an alternative DA approach ("off-line" or "no-cycling") was applied by several scholars (Okazaki and Yoshimura, 2017; Dee et al., 2016; Acevedo et al., 2017; Steiger and Hakim, 2015; Chen et al., 2015; Steiger et al., 2014). In an offline DA, the model initialization from the analysis has been abandoned and the background ensembles are derived from precomputed simulations (Okazaki and Yoshimura, 2017). In this paper we conduct a test by assimilating yearly averaged pesudo and real observations into an ensemble of RCM simulations over Europe via an offline DA approach. The main purpose of our study is to contribute to the simulation of the climate in general, and in particular, to the paleo-DA efforts, by addressing following points:

(i) How is the geographic distribution of model bias in an ensemble of regional climate simulations (members differ slightly in boundary and initial conditions)?

(ii) What is the optimum radius within which the observations are truly correlated?

(iii) Can this particular offline DA approach be used to constrain the regional climate simulations in time and space?

(iv) What is the range of observation errors within which the observations reduce the model's bias via DA?

This paper is structured as follows. Section 2 starts with introducing the offline DA basics and the experiment design. Furthermore, the concept of the perfect model experiment and the metrics, which measure the performance of this method are described. In Sec.3 we present our results of constrained model simulations. We discuss and summarize our work in Sec.4.

## 2   Data and Methods

### 2.1   Optimal Interpolation Basics

Prior to describing the experimental design, we give a brief review of the optimal interpolation (OI) method (for the full review see Barth et al. (2008a, b)). OI or "objective analysis" or "kriging" is one of the most commonly used and simple Data Assimilation (DA) methods applied since 1970s (Barth et al., 2008b). The unknown state of the climate $X$ is a vector, which





has to be estimated conditioned on the available observations ($Y$). Given the state vector $X$, the state on observations' location is obtained by an interpolation method (here nearest neighbor). This operation is noted as matrix $H$ (observation operator) and consequently the state at the observations' location is defined as $HX$. If the observations are in the form of proxy data, the operator $H$ will bring the model state to the proxy state by means of a proxy forward model. The ultimate goal of the OI is to

5 estimate the "True" state of the climate ($X^{TRUE}$) the so-called analysis ($X^A$) given the observation ($Y$) and background $X^B$ (first guess). The background and observation can be written as :

$$X^B = X^{TRUE} + \eta^B \tag{1}$$

$$Y = HX^{TRUE} + \varepsilon \tag{2}$$

where $\varepsilon$ and $\eta^B$ denote the observation and background errors, respectively. In the applied OI scheme here, it is assumed that the background and observations are unbiased:

$$E[\eta^B] = 0 \tag{3}$$

$$E[\varepsilon] = 0 \tag{4}$$

Other hypotheses are that the information about the observation and background errors are known (*prior* knowledge) and they are independent:

$$E[\eta^B \eta^{B^T}] = P^B \tag{5}$$

$$E[\varepsilon \varepsilon^T] = R \tag{6}$$

$$E[\eta^B \varepsilon^T] = 0 \tag{7}$$

The Optimal Interpolation (OI) scheme is considered as the Best Linear Unbiased Estimator (BLUE) of the true state $X^{TRUE}$. A BLUE has the following characteristics:

1. It is linear for $Y$ and $X^B$





2. It is not biased:

$$E[X^{Analysis}] = X^{NATURE} \tag{8}$$

3. It has the lowest error variance (optimal error variance).

The unbiased linear equation between $X^B$ and $Y$ can be written as :

$$X^{Analysis} = X^B + K(Y - HX^B) \tag{9}$$

where K is the "Kalman gain" matrix. Equation 9 can be written as :

$$\eta^{Analysis} = \eta^B + K(\varepsilon - H\eta^B) = (I - KH)\eta^B + K\varepsilon \tag{10}$$

Thus the error covariance of the analysis will be:

$$P^{Analsis}(K) = E[\eta^{Analysis}\eta^{Analysis^T}] = (I - KH)P^B(I - KH)^T + KRK^T \tag{11}$$

The trace of matrix $P^{Analsis}$ indicates the error covariance of the analysis:

$$Trace(P^{Analsis}(K)) = Trace(P^B) + Trace(KHP^BH^TK^T)$$
$$-2Trace(P^BH^TK^T) + Trace(KRK^T) \tag{12}$$

Given that the total error variance of analysis has its minimum value, a small $\delta K$ will not modify the total variance:

$$Trace(P^{Analsis}(K + \delta K)) - Trace(P^{Analysis}(K)) = 0$$
$$= 2Trace(KHP^BH^T\delta K^T) - 2Trace(P^BH^T\delta K^T) + 2Trace(KR\delta K^T) \tag{13}$$
$$= 2Trace([K(HP^BH^T + R)]\delta K^T)$$

Assuming that the $\delta K$ is arbitrary, the Kalman gain is :

$$K = P^BH^T(HP^BH^T + R)^{-1} \tag{14}$$

Finally, the error covariance of the BLUE is given by:

$$P^{Analysis} = P^B - KHP^B$$
$$= P^B - P^BH^T(HP^BH^T + R)^{-1}HP^B \tag{15}$$

The calculation of the covariance matrices for RCM are very expensive. Therefore, an ensemble of the model states are applied to approximate the mean and covariance of the forecast (Evensen, 1994). Following a stochastic approach (Hamill, 20    2006), an observation ensemble is created by adding random noise (in the observational range) to the $Y$. In our experiment, we tested the impact of different observation errors on the analysis skill in more detail.



## 2.2 Observation System Simulation Experiment

Models contain systematic errors which may have diverse origins (dynamical core, parametrization and initialization). DA schemes are also based on simplified hypotheses and are imperfect (e.g., here the Gaussian parametrization for $P^B$). The interaction of error sources with one another obscures tracing of the origins of such biases. These caveats are neglected by
using a simplified numerical experiment called observation system simulation experiment (OSSE). The usage of OSSEs is increasing in the field of climate DA as a validation tool (Annan and Hargreaves, 2012; Bhend et al., 2012; Steiger et al., 2014; Acevedo et al., 2015; Dee et al., 2016; Acevedo et al., 2017; Okazaki and Yoshimura, 2017). Firstly, the nature model simulation $X^{NATURE}$ ("true" run) is conducted as the prediction target. Then by using the output from the nature run and by adding random draws from a white noise distribution, the pseudo-observations are created which are interpolated over the
observations' location. The location of 500 random meteorological stations of the "ENSEMBLES daily gridded observational dataset for precipitation, temperature and sea level pressure in Europe called E-OBS" (Haylock et al., 2008) are used to create the pseudo-observation data (Fig.1). Finally, the OI scheme is applied to assimilate the pseudo-observation into the free ensemble run and the observationally constrained run $X^{DA}$ is obtained. Here we neglect the reinitialization step of the DA and draw the forecast state from precomputed free ensemble simulation. Recently such assimilation methodologies are labeled as
"off-line" DA (Huntley and Hakim, 2010).

## 2.3 Ensemble Generation Technique

RCM simulations tend to follow the trajectory of the driving GCM, however it is known that RCM deviate from the driving GCM, both on smaller scales, which are not resolved by the GCM and on larger scales which are resolved by the GCM (Becker et al., 2015). RCMs react very sensitive to the chosen domain and usually are tuned for the selected area. Modifying the RCM's
domain in size or geographic area will set new boundary conditions for the RCM. In the context of the domain manipulation, several studies analyzed the impact of the domain size on regional model simulations and find significant impact of the domain size on predictive skill and climatological characteristics of the models (Larsen et al., 2013; Goswami et al., 2012; Colin et al., 2010). In addition, a shift of the model domain generates different large-scale patterns in the simulation outputs (Miguez-Macho et al., 2004), which are the consequence of the "large-scale secondary circulations" in the RCM and are relative to the
driving data (Becker et al., 2015). There are different popular methods for generating RCM ensembles, e.g. the use of different parameterizations (Wang et al., 2011), different driving data (Verbunt et al., 2007) or different initialization dates (Hollweg et al., 2008). However, the uncertainty introduced by the selection of a specific model domain is rarely considered, but the usefulness of this approach is shown in Pardowitz et al. (2016) and Mazza et al. (2017). The advantage of this method is that it is easy to implement and allows a consistence model setup among the ensemble members. The multiple members are created
by shifting the model domains relative to each other.





## 2.4 Experimental Design

The CCLM model version cosmo_131108_5.00_clm8 (Asharaf et al., 2012) is used as the dynamical model and the nearest neighbor interpolation is applied as the observation forward model. We have to note that the model-data mapping is still a limiting factor in paleo DA (Dee et al., 2016; Goosse, 2016). For a review of proxy system models we refer to the recent work of Dee et al. (2016). Here, our focus is only on the DA scheme and its usability in paleo-climate. Two sets of observations are used by time-averaging of daily values over the winter (DJF) and summer (JJA) seasons. The time average climate analysis is conducted by using the OI scheme as the DA tool. The horizontal resolution of the simulations is set to $0.44° \times 0.44°$. The full model set-up file is accessible online (ref. 4). Our OSSE consists of two sets of simulations:

- **Nature** simulation: a 10 year long simulation over Europe driven by global atmospheric reanalysis data (6-hourly) produced by the European Centre for Medium-Range Weather Forecasts (ECMWF), the so-called ERAInterim (Dee et al., 2011) (initial and boundary conditions are taken from ERAInterim). This run will be used as the "target" state of the climate in the investigations and is labeled as the "nature" run.

- **Shifted** domain simulations: all the model parameters are set as in the nature run but simulation domains are shifted in 4 distinct directions (northeast, northwest, southeast and southwest) and with 5 different shifting values (1 to 5 grid points) from the nature domain. These shifting directions allow for having optimal different boundaries compared to the nature run (Fig. 1). Due to high computational expenses of such simulations we conducted 10-year simulations (total number of $21 \times 10 = 210$ model years).

## 2.5 Model's Skill Metric

The skill of the free ensemble run can be quantified by the root mean square error (RMSE):

$$RMSE(\langle X^{Free} \rangle) = \left( \overline{ \left( X^{Nature} - \langle X^{Free} \rangle \right)^2 } \right)^{\frac{1}{2}},$$

(16)

where $\overline{\phantom{x}}$ and $\langle \, \rangle$ denote the time and ensemble mean operator. Mean and spread of the free ensemble are presented as :

$$\langle X_{free} \rangle = \frac{1}{N} \sum_{n=1}^{N} x_n^{free},$$

(17)

$$\sigma(i,j) = \sqrt{ \frac{1}{N-1} \sum_{n=1}^{N} (\epsilon_n(i,j))^2 }$$

(18)

where $\epsilon_n(i,j) = X_m(i,j) - \overline{\langle X \rangle}(i,j)$, $N$ is the ensemble index and $(i,j)$ the indices of horizontal positions. Same approach is used to calculate the skill for the analysis quantities.



To maintain clarity we focus only on the temperature at 2 meter (T2M) variable. For evaluation, the relaxation zone where the boundary data is relaxed in the RCM (here 20 grid points next to the lateral boundaries) is removed. The sea surface temperature in our CCLM simulation is interpolated from the driving model (here ERAInterim) and not calculated by the model dynamics and the spread of the ensemble for T2M is zero over oceans. Therefore the RMSEs over oceans are masked

out from the analysis and values over land are only shown.

## 3 Results

### 3.1 Unconstrained Ensemble Runs

Figure 2 shows the seasonal mean of RMSE and spread of the ensemble for T2M over the evaluation domain. A slight change in the boundary conditions (shifting of the domain) of the model leads to large RMSEs (up to $1K$ in seasonal means) in

the forecasted quantities (Fig.2). Maximum RMSE values are located on the regions where the spread of the ensemble is also large. An interesting feature of the RMSE pattern is the accumulation of the errors in the center and Northeast of the domain for both the winter and summer seasons. West and North of the domain is largely dominated by the ocean where the temperature quantities are forced by the reanalysis data. There is small difference between the ensemble members on these areas and the maximum error values are mostly located on land. On the other hand, the maximum lateral inflow occurs on the West and the

Northwest boundaries. These features have to be considered cautiously when conducting long-term climate simulations using CCLM. The long seasonal time averaging filter relatively dampens the RMSE and the spread of the ensemble. The errors will increase drastically for daily quantities. One of our main motivations in this paper is to test the usage of OI in calculation of analysis quantities by assimilating the pseudo and real observations in the free ensemble simulations.

### 3.2 Constrained Ensemble Runs

The error covariance of the BLUE, $P^B$ can be localized through the following parametrization:

$$P^B(x_1,...,x_n,y_1,...,y_n) = \sigma(x_1,...,x_n)^2 exp(-\frac{(x_1-y_1)^2}{L^2{}_1})...-\frac{(x_n-y_n)^2}{L^2{}_n})) \qquad (19)$$

where the $\sigma(x_1,...,x_n)^2$ is the error variance and $L_n$ the correlation length (or "localization radius"). Background state on each grid point may be highly correlated with observations over regions far apart. The correlation length is defined to overcome these spurious relations. Prior to conducting the analysis calculation, we calculate the optimal correlation length ($L$) for the

covariance localization. The mean RMSE over the evaluation domain is calculated for different values of $L$ ($0.1° \leq L \leq 6°$). The RMSE values show a minimum at correlation length of $1.7°$ ($\sim 190\ Km$) for summer (JJA) and $2.1°$ ($\sim 230\ Km$) for winter (DJF) (Fig.3). For a long-term time averaging e.g., yearly or decadal periods, the correlation length will increase accordingly (Chen et al., 2015). The increase rate of the RMSE with respect to the correlation length is larger during the summer than in winter.

The error reduction of the DA is influenced by the shape of the white noise added to create the pseudo-observation data. Here, we assess the noise levels by signal-to-noise ratio (SNR), which is the ratio of the variance of clean observation to the





variance of added white noise. Figure 4, shows the averaged seasonal near-surface temperature RMSE for the analysis of the ensemble run. The RMSE values are reduced linearly with increasing SNR. In the study of Acevedo et al. (2017), the RMSE values did not change significantly between 2 and 11 SNR values. One possible reason for such a behavior might be due to the regional domain of RCM over Europe, which is dominated by local effects. On the other hand, $\sim 70\%$ of the GCM domain in

Acevedo et al. (2017) was covered by the oceans where no proxy was assimilated. In our experiment, the field-mean filter does not flatten the RMSE reduction rate for larger SNR values over Europe. For the rest of the paper the results are shown with $SNR = 3$.

Figures 5 and 6 show the error reduction of analysis for summer and winter. Assimilation of 500 pseudo-observations has significantly reduced the error for both seasons. However, the spread of the errors is highly reduced in summer than in winter.

The regions with the largest error reduction are located at highly populated areas by pseudo-observations (e.g., Germany, Sweden and Denmark). One interesting feature is the error reduction over the South of the domain where only 3 observations are available. Usually, in an offline DA the observational information would not be accumulated over time and could not be conveyed to unobserved regions (Acevedo, 2015). However, there exists a universal behavior in the fluctuations of the terrestrial near surface temperature (Fallah et al., 2016a), which might be the reason of propagation of the information to the other areas.

**3.3  Long Ensemble Runs**

As defined by world meteorological organization (WMO) the climate is explained by averaging the weather state for a period of at least 30 years. Therefore, we conducted a new set of 5 36-year simulations (one nature and 4 shifted runs). The computational costs of RCM was the only limiting factor to choose this number of members ($5 \times 36 = 180$ years of model run). The members were perturbed by shifting the domain 4 grid points to the Northeast, Northwest, Southwest and Southeast.

The 36-year ensemble spread and RMSE show similar patterns as in the previous experiment using 20 members (Fig. 7). The analysis quantities indicate a significant error reduction both in median and spread of the ensemble (Fig. 8 and Fig. 9). Figure 10 illustrates time evolution of field-mean RMSEs for the free ensemble and analysis quantities. There exists a linear trend in free runs' RMSE. However, the linear trend is removed in the RMSE of the analysis for the summer. This feature was previously observed in an offline DA experiment using an intermediate complexity model and the EnKF approach (Acevedo

et al., 2017). Furthermore, the results show that the analysis RMSEs are significantly reduced compared to the free ensemble run.

**3.4  Assimilation of E-OBS Observations**

A DA test with the real observation data will shed light on the model behavior and efficiency. It should be mentioned that the interpolation of station data to grid structure of E-OBS data set is done using the kriging method for temperature quantities

(ref. Haylock et al. (2008)). Time-averaged station observations from E-OBS are assimilated into a 36-year ensemble of 4 CCLM simulations. For comparison of model and gridded E-OBS dataset at $0.44°$, the latter is bi-linearly remapped on the CCLM grid structure. Figures 11a and 11c show the RMSE of the ensemble mean with respect to E-OBS data set for winter and summer. Regions with low model bias are located over East and North Europe: Germany, France, Spain, Portugal, Finland and



United Kingdom for both winter and summer. In winter large biases are located over Switzerland, Italy, Morocco, Algeria and Russia. The RMSE values are generally larger in summer with high biases over Southeast Europe and around Mediterranean Sea for example Bulgaria, Greece, Albania, Serbia and Ukraine. Figures 11b and 11d show that the model bias is largely reduced by assimilating 500 E-OBS observations. Figure 12 displays the significance of the mean error reduction after the

assimilation. For both seasons the RMSEs median are significantly reduced with respect to the unconstrained runs.

The importance of a homogeneous spatially distributed observation network is highlighted in the video clips in Appendix A. By increasing the number of assimilated observations to 2700 the RMSEs are largely reduced. This experiment shows that even a small number of stations (100) can contribute to a large error reduction in the analysis quantities. The distribution pattern of E-OBS stations in Central Europe is denser than East, West and South Europe. Therefore, areas with coarse station population

might require larger number of stations in order to achieve a homogeneous error reduction in the analysis.

## 4   Conclusions

Using a computationally fast DA approach, we assimilated pseudo and real observations within an ensemble of precomputed RCM simulations. The ensemble is created by slightly perturbing the boundary and initial conditions of its members via the domain shifting method. In the framework of a perfect model experiment the performance of free ensemble and analysis

quantities is evaluated. Such experiments facilitate the estimation of observation, background and analysis error. In the first set of experiments, we conducted an ensemble of 20 simulations driven by ERAInterim for the duration of 10 years. The nearest-neighbor interpolation is applied as the observation operator plus a random white noise with known standard deviation to create a set of pseudo-observations from the nature run. Pseudo-observations are assimilated within the ensemble of RCM runs by means of the OI approach. By conducting a set of simulations using 4 perturbed members and a nature run, we repeated the

perfect model experiment for a time expansion of 36 years. This allowed us to draw conclusions on the time evolution of the DA skill for a typical climatological period (more than 30 years). In a final step, we assimilated real observations from E-OBS in an ensemble of 4 RCM simulations of 36 years duration and compared the results with the E-OBS gridded data set at 0.44° horizontal resolution.

The comparison of ensemble mean of COSMO-CLM model outputs and the pseudo-observations shows that the model

seems to be well tuned for Central Europe. A region of significant model bias for both winter and summer seasons is located over the East side of the domain. This area is located far from the ocean where the ERAInterim data is prescribed (no coupled ocean was implemented). Therefore, we speculate that the model generates more variabilities and is free to evolve over this region. This biased behavior is also observed in a real DA experiment using the E-OBS observation data set (answer to question (i) in Introduction). Furthermore, we iterated the DA experiment on different values of correlation length for the summer and

winter to find the optimal correlation length quantity. The optimum radius of correlation is found to be $1.7°$ ($\sim 190 \ Km$) for summer (JJA) and $2.1°$ ($\sim 230 \ Km$) for winter (answer to question (ii) in Introduction). Afterward, we showed that the skill of OI is linearly influenced by the SNRs used to create the noisy observations (answer to question (iv) in Introduction).



Our experiments showed that the ensemble OI is useful for conducting analysis of seasonally averaged quantities (answer to question (iii) in Introduction). Despite inhomogeneity of observation distribution over the domain, the analysis presents error reduction over most of the domain. For a small ensemble with longer integration period of 36 years, the analysis significantly outperforms the ensemble mean. However, for the winter season, the analysis error is increasing with time and it consists the same rising trend as in the error of the ensemble mean. For the summer time the trend was removed in the analysis. This was previously observed in the study of Acevedo et al. (2017) by assimilating the summer-time tree ring width pseudo observations in an AGCM using the EnKF. However, our simulations are very short compared to the one of Acevedo et al. (2017). The disability of OI in removing the winter-time RMSE's trend might be improved by applying an offline EnKF scheme as used by Okazaki and Yoshimura (2017), Acevedo et al. (2017) and Dee et al. (2016). Using a random climatology from a large climate simulation pool as the background instead of using an ensemble simulation at the observation period, following Steiger et al. (2014) and Dee et al. (2016), might remove the winter RMSE trend. However, the magnitude of the analysis skills might be influenced by using the climate of a single run for the background state. By now available long RCM runs of this study, Russo and Cubasch (2016), Fallah et al. (2016b) and the Coordinated Downscaling Experiment-European Domain (EURO-CORDEX), which can serve as a large climate analog for the background state, we suggest further DA experiments to reconstruct high resolution climate fields given the time-averaged observations.

A major culprit of our experiment is linearity assumption of the forecast model and Gaussianity of the observation and model errors. In OI the background error covariance is usually prescribed and calculated once during the entire assimilation procedure. Our experiment showed that although the spread of the ensemble is increasing slightly in time, each individual member and the ensemble mean show a similar trend. This similar behavior of the members might be due to the systematic behavior of the CCLM. We suggest a multi-model ensemble approach to account for a wider range of internal variabilities. However, conducting such experiments are prohibitively expensive with today's computational powers. In a real world experiment, it is recommended to use the proxy system models (PSMs) to remove the simplistic assumptions of inverse climate modeling and let the assimilation to take place at the observation space instead of model space (Dee et al., 2016). However, there is a huge gap in the recent knowledge of PSMs and more efforts shall be made to move the knowledge of forward proxy modeling one step forward (Acevedo et al., 2017).

*Code and data availability.* The experiment code and its full description are available at: https://github.com/bijanfallah/OI_CCLM. A test set of climate model simulations for temperature quantities is available at http://users.met.fu-berlin.de/~BijanFallah/NETCDFS_CCLM/. The full model output can be available upon request by BF. The OI code and its description is public at http://modb.oce.ulg.ac.be/mediawiki/index.php/Optimal_interpolation_Fortran_module_with_Octave_interface by GeoHydrodynamics and Environment Research (GHER) research group.



## Appendix A

Maps of time-averaged T2M RMSEs after assimilation of real observations with different number of stations for summer and winter are shown in the movies within assets. The number on South-West corner of the frame indicates the number of observations, which are assimilated.

5  *Author contributions.* BF and WA designed the experiment. BF has done the scientific coding and performed the simulations. All authors analyzed the results and drafted the manuscript.

*Acknowledgements.* The authors gratefully acknowledge German Federal Ministry of Education and Research (BMBF) as Research for Sustainability initiative (FONA); through PalMod, a German-funded (DFG) project with the general goal of analyzing a complete glacial cycle, from the Last Interglacial to the Anthropocene. The computational resources were made available by the German Climate Computing
10  Center (DKRZ). We acknowledge GeoHydrodynamics and Environment Research (GHER) research group for making their OI code available for public. Our special thanks go to Ingo Kirchner for his helpful comments on simulation set-up as well as the CCLM community.



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





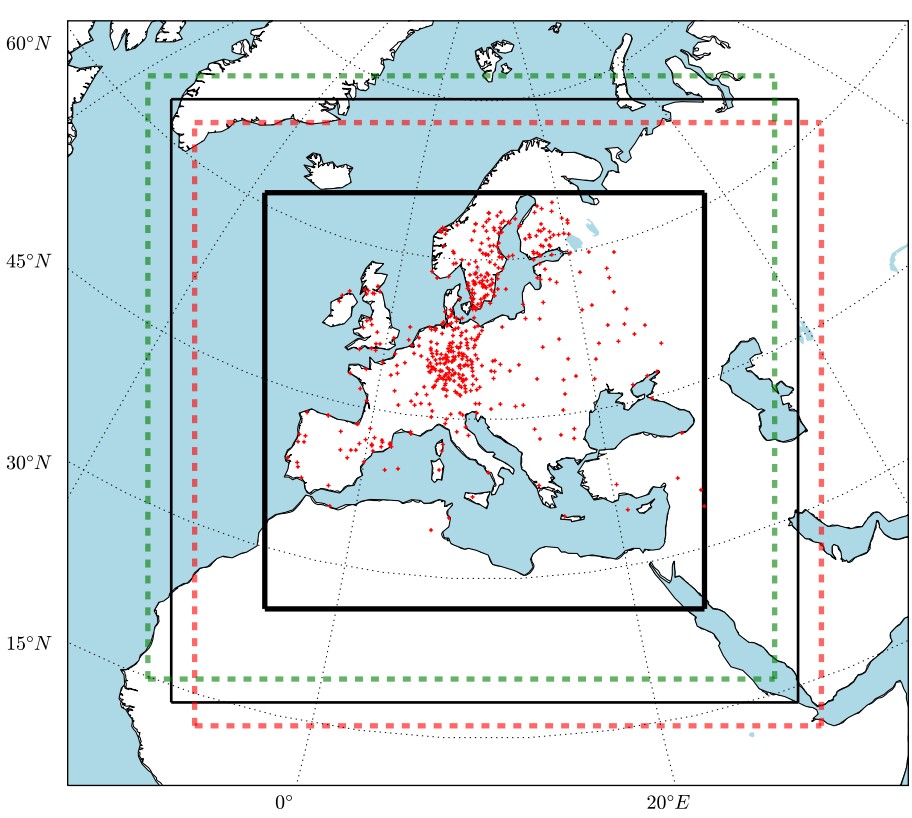

**Figure 1.** Regional Climate Model (RCM) domains: black thin box shows the nature and dashed boxes show the shifted members (only two are shown for Northwest and Southeast with 5 grid points distance from nature) and red pluses show 500 random meteorological stations of the "ENSEMBLES daily gridded observational dataset for precipitation, temperature and sea level pressure in Europe called E-OBS" (Haylock et al., 2008). The thick black box shows the evaluation domain.





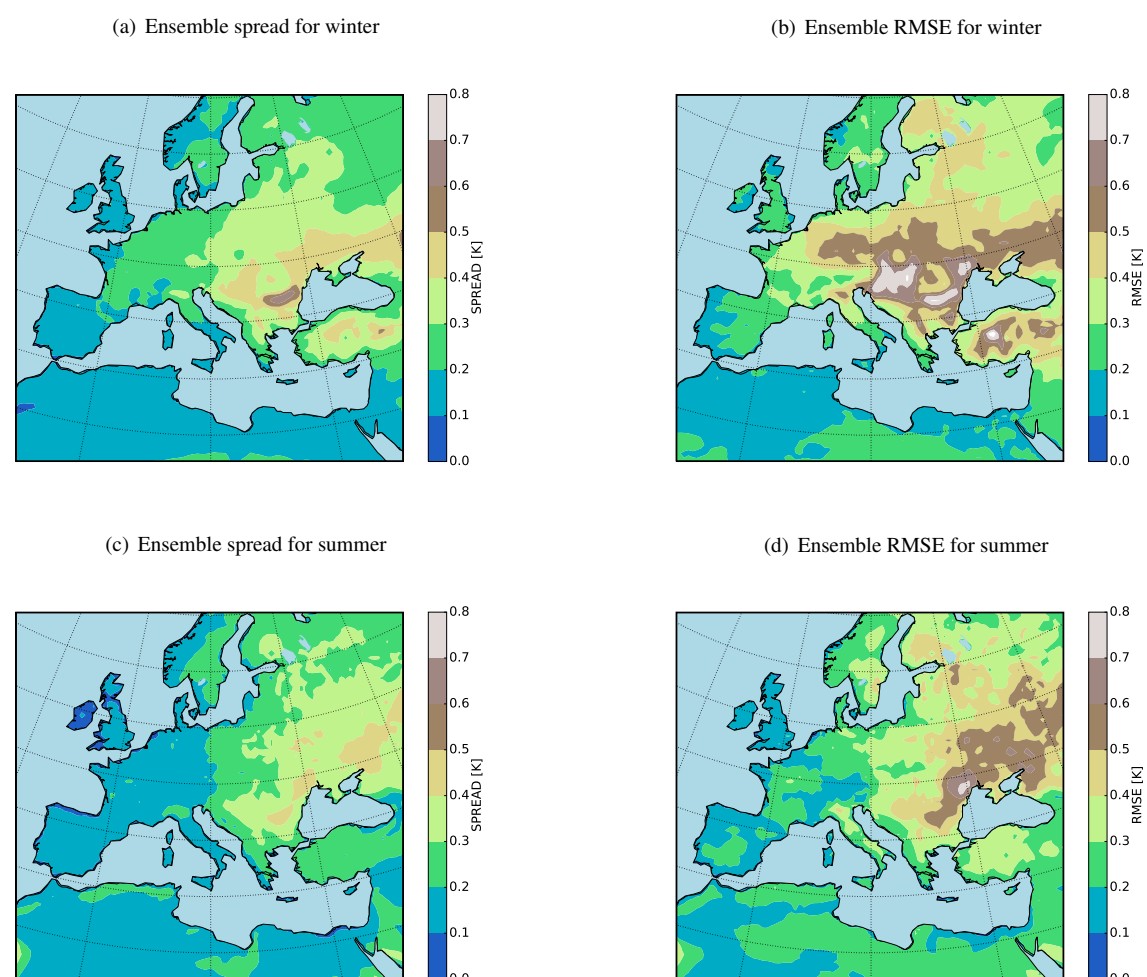

**Figure 2.** 10-year ensemble spread and RMSE for seasonal mean of 2 meter temperature (K).





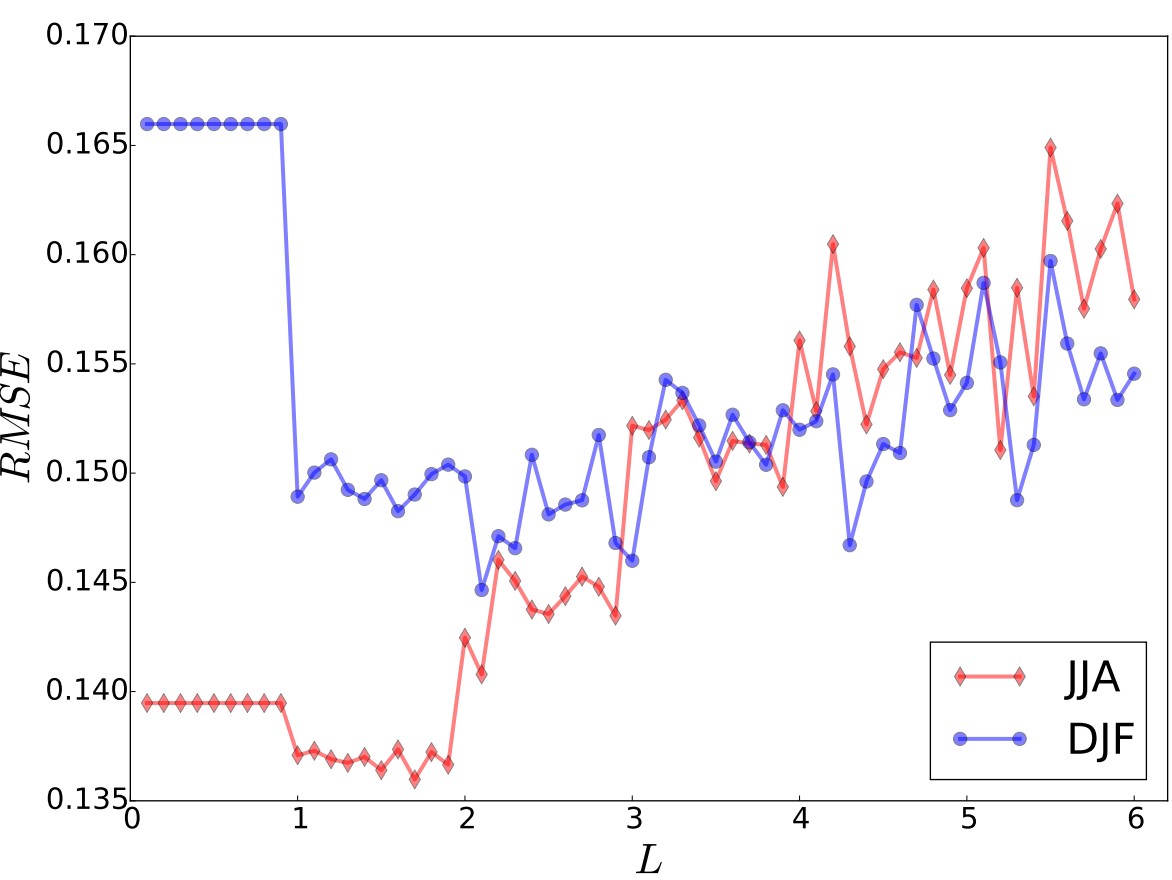

**Figure 3.** Field-mean of RMSE for near surface temperature ($K$) analysis over the evaluation domain for different correlation length ($L[°]$) for Winter (blue) and Summer (red).





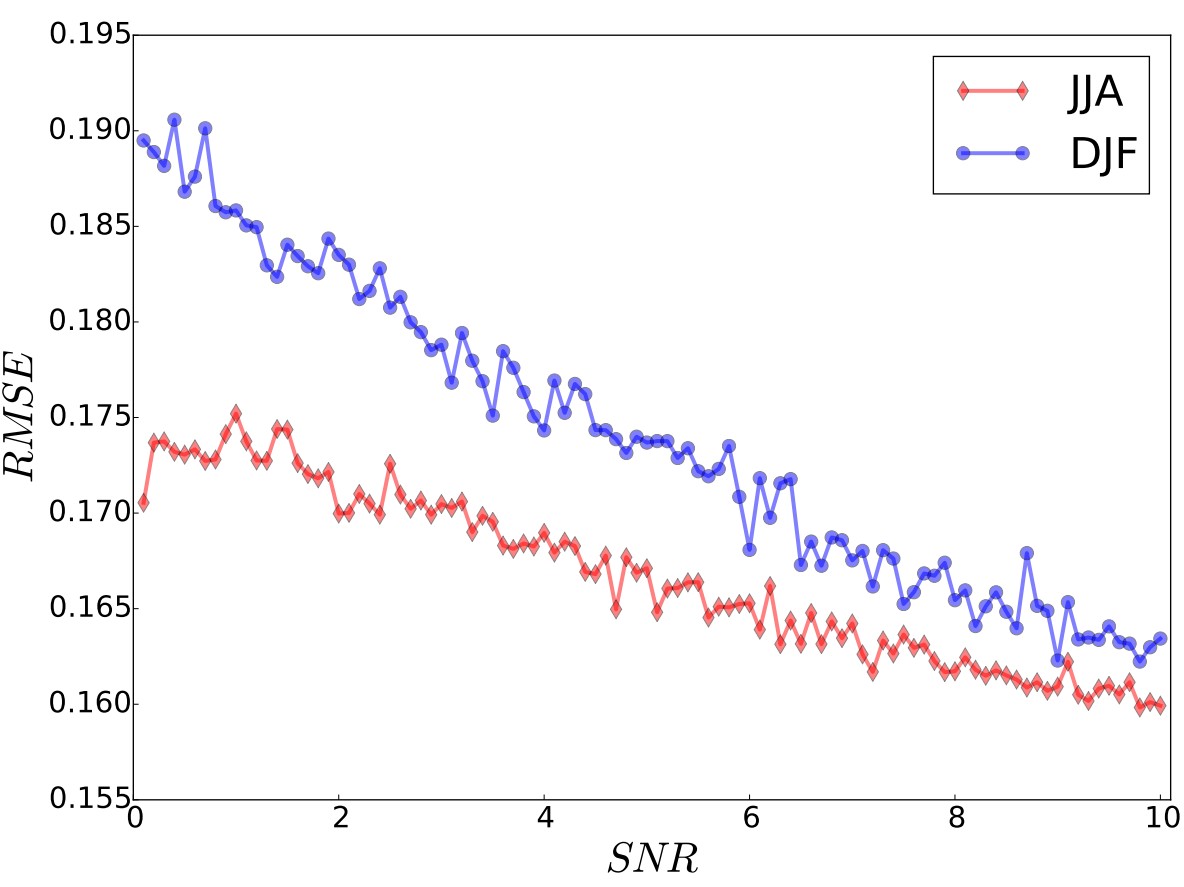

**Figure 4.** Field-mean of seasonal near-surface temperature RMSE [K] of analysis vs SNR.





(a)

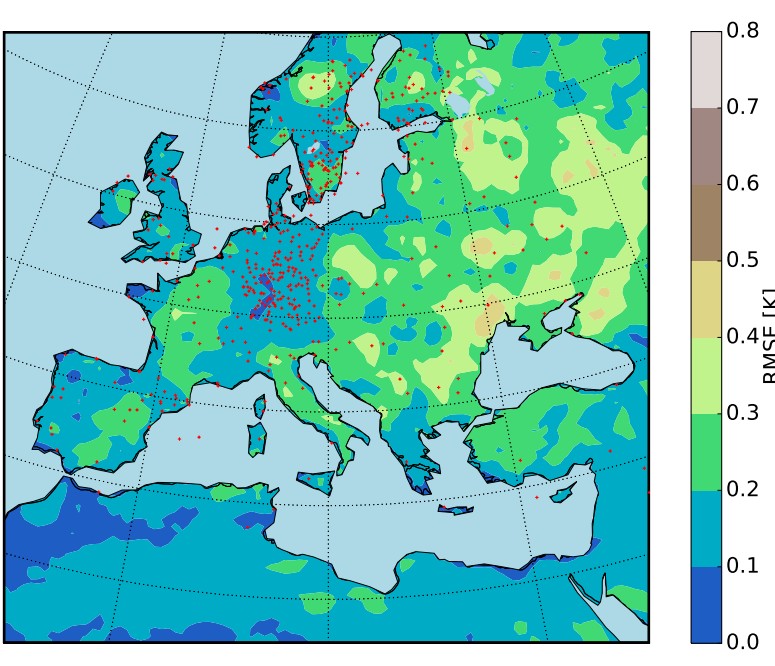

(b)

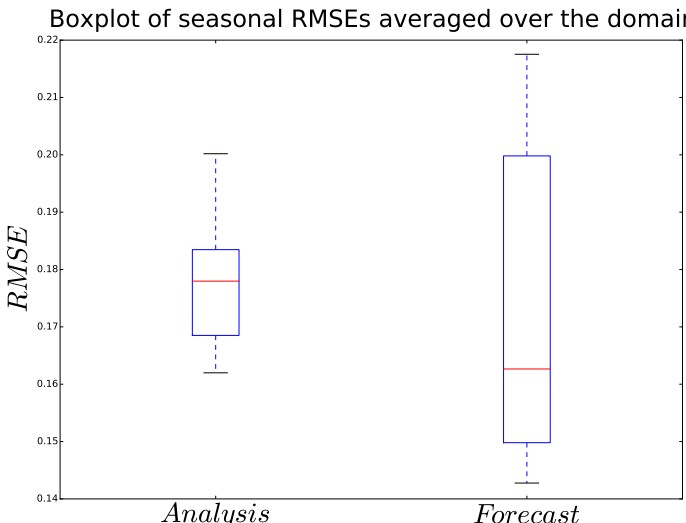

**Figure 5. (a)**: 10-year RMSE [K] of analysis for summer (JJA) and **(b)**: the field-mean RMSE [K] of analysis and free ensemble run for summer (JJA).



(a)

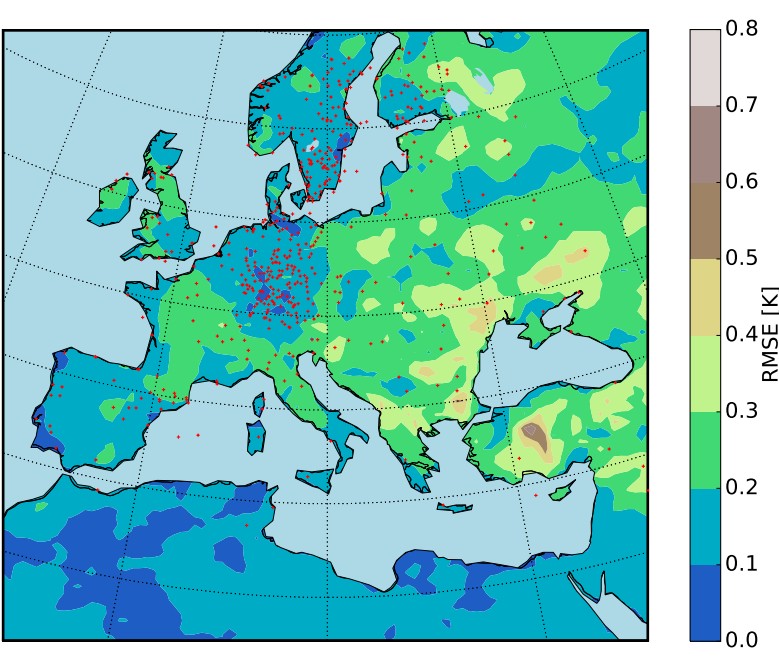

(b)

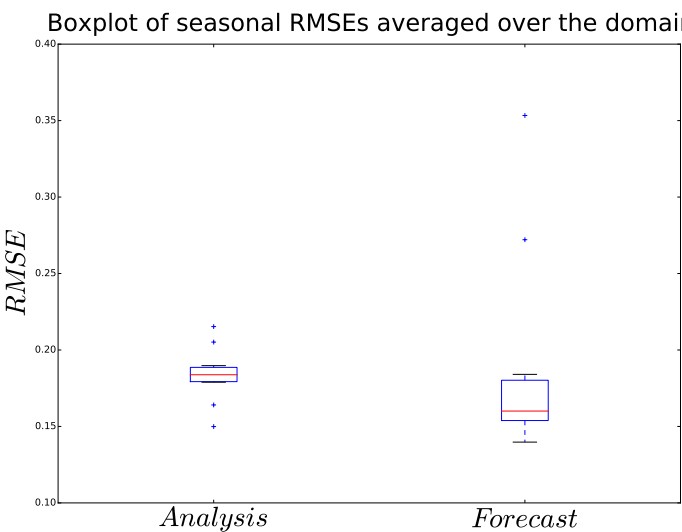

**Figure 6. (a)**: 10-year RMSE [K] of analysis for winter (DJF) with and **(b)**: the field-mean RMSE [K] of analysis and free ensemble run for winter (DJF).





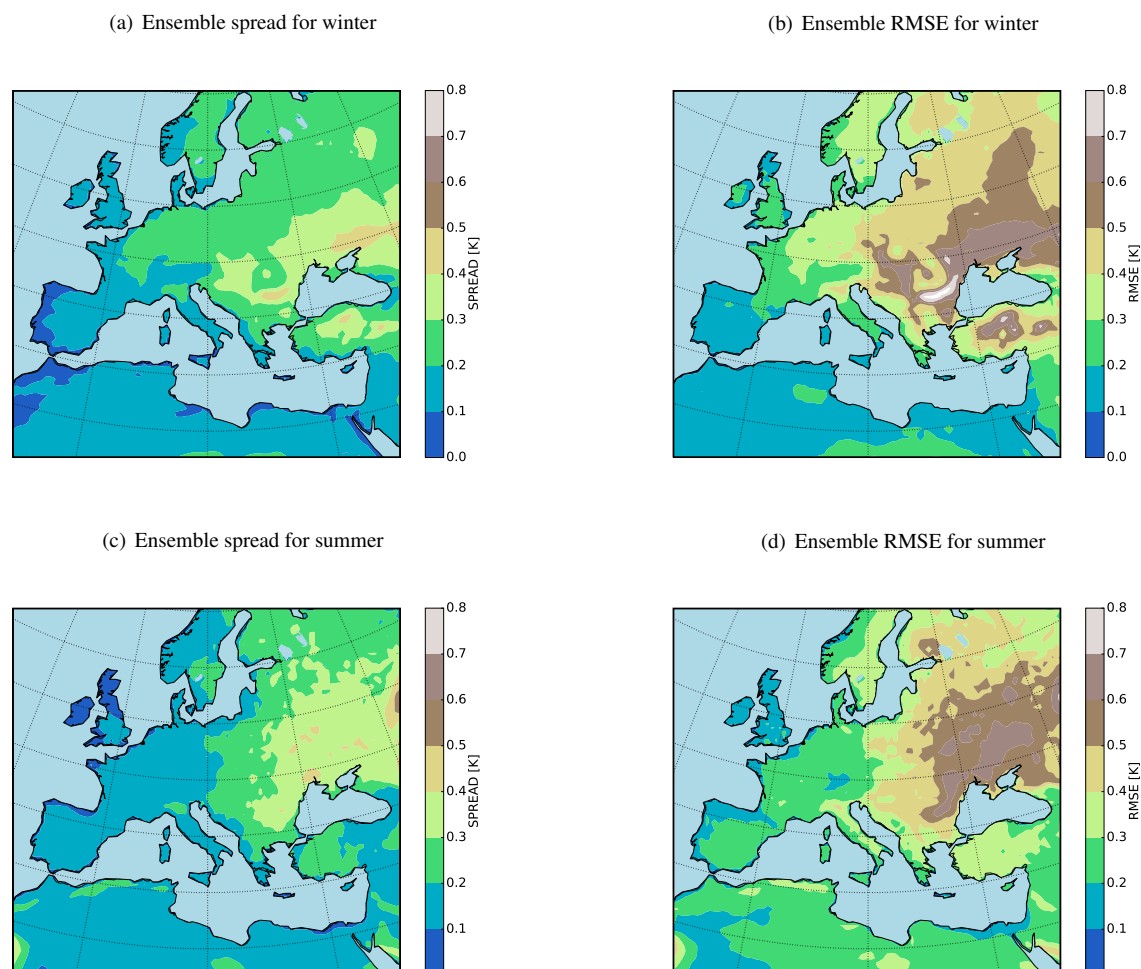

**Figure 7.** 36-year ensemble spread and RMSE for seasonal mean of 2 meter temperature (K).





(a)

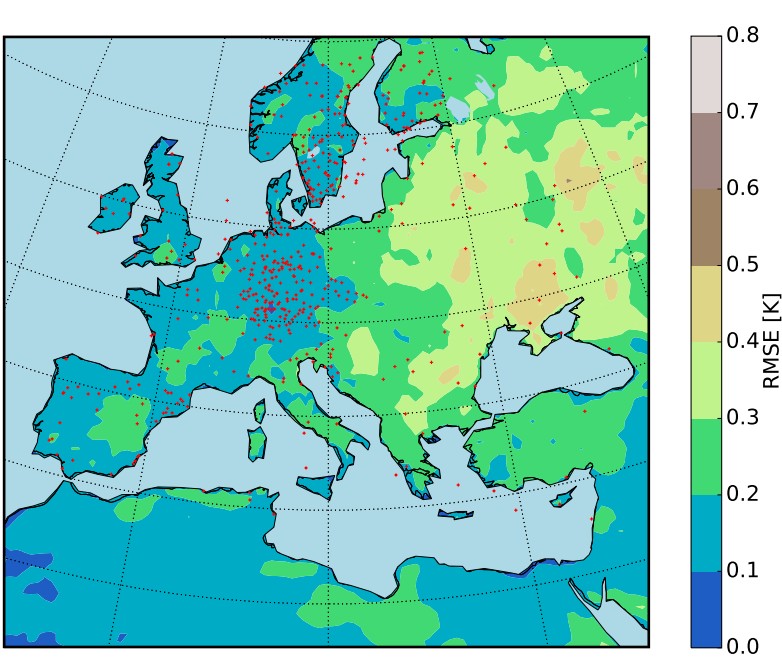

(b)

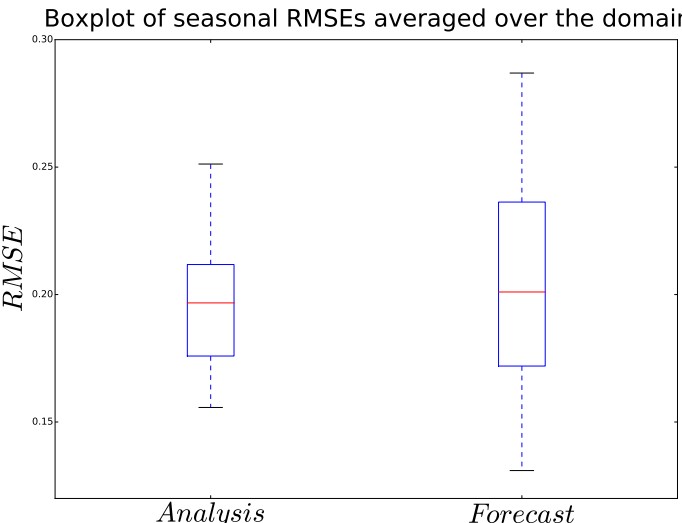

**Figure 8. (a)**: 36-year RMSE [K] of analysis for summer (JJA) and **(b)**: the field-mean RMSE [K] of analysis and free ensemble for summer (JJA). Red squares indicate the mean of RMSEs.



(a)

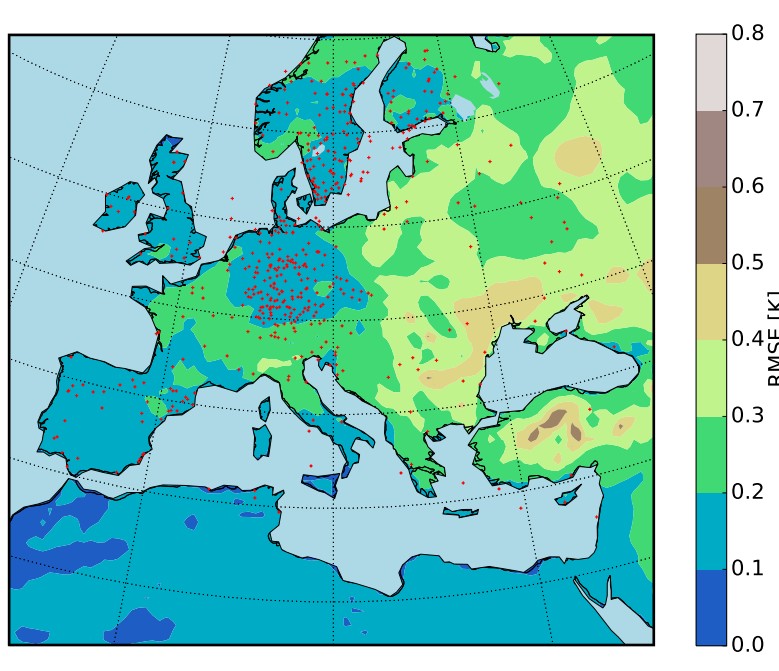

(b)

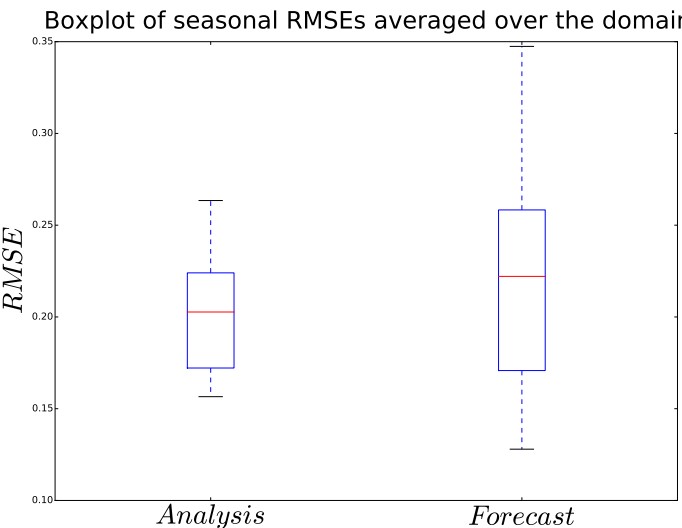

**Figure 9. (a)**: 36-year RMSE [K] of analysis for winter (DJF) and **(b)**: the field-mean RMSE [K] of analysis and free ensemble run for winter (DJF). Red squares indicate the mean of RMSEs.







**Figure 10.** Yearly field-mean RMSE [K] of ensemble mean (white line) and analysis (black line) for winter **(a)** and for summer **(b)**. Shadings show the ensemble members and dashed lines indicate the linear trends.



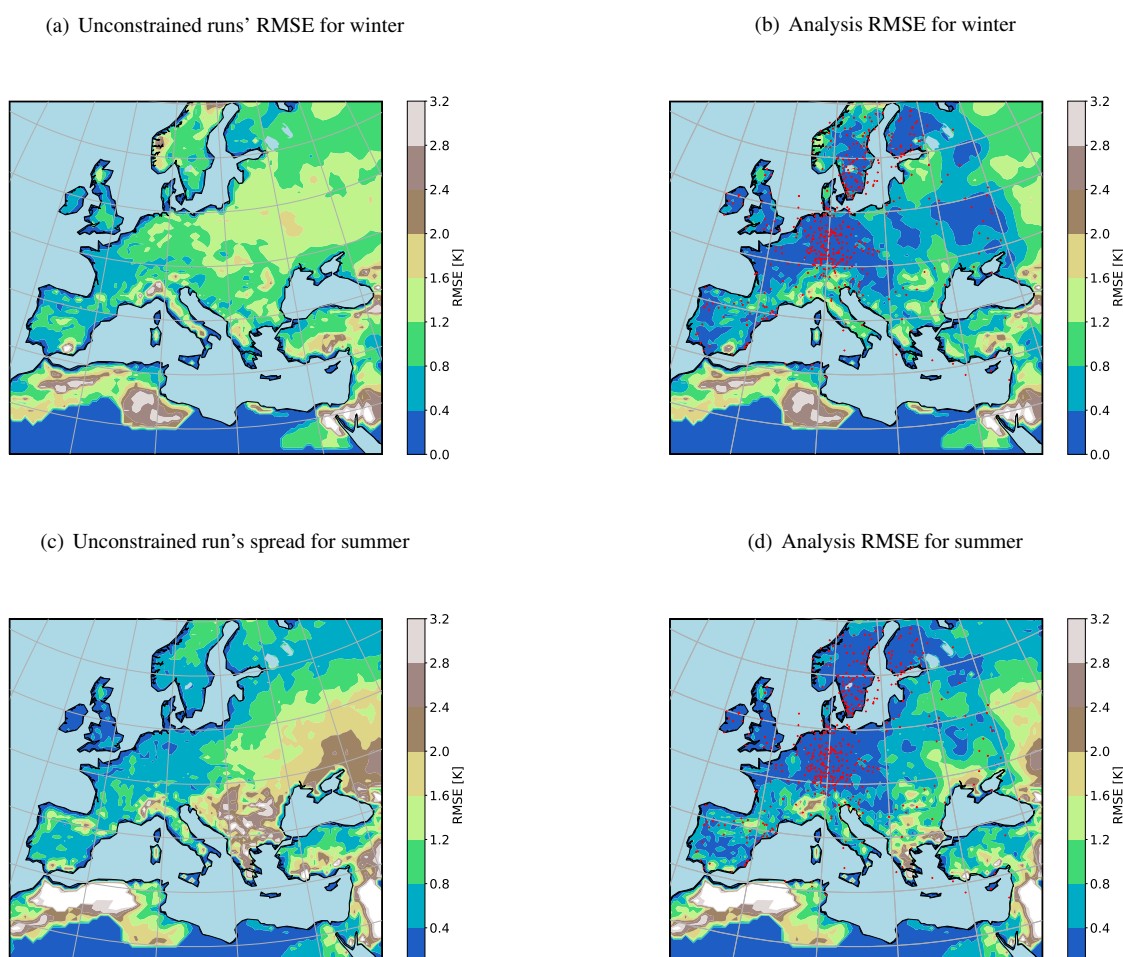

**Figure 11.** 36-year unconstrained runs' **(a,c)** and analysis' RMSE **(b,d)** for seasonal mean of 2 meter temperature (K) with respect to E-OBS data.. Red crosses show the location of 500 selected observations from E-OBS.





**Figure 12.** Yearly field-mean RMSE [K] of ensemble mean (white lines) and analysis (black lines) for winter **(a)** and for summer **(c)** along with their box plots **(b,d)** with respect to E-OBS data. Shadings show the ensemble members and dashed lines indicate the linear trends.