# Peer review of "Towards High Resolution Climate Reconstruction Using an Off-line Data Assimilation and COSMO-CLM 5.00 Model"

_Climate of the Past, 2017_

## Referee Comment (RC1) · Anonymous Referee #1 · 30 Nov 2017

The paper uses the Ensemble Optimal Interpolation (EOI) method with time-averaged observation values to reconstruct the climate states by means of high resolution CCLM simulations, which are driven by the ERAInterim. The authors set up 3 different test runs to go through different aspects of the EOI. They show that the analysis has significant skill compared to the free ensemble simulations. The authors apply an offline method which is widely used in the paleo-DA community and have been shown successful for at least Climate Field Reconstruction (CFR). They show that the error reduction of the EOI is linearly related to the observation error. The word "towards" in the title confirms that the paleo-DA is still in its infancy and has to be discovered deeper.

Overall, it is a nice study that can impact the climate field reconstruction efforts and the results are scientifically sound. The techniques might be used by further studies for the high resolution paleo-climate reconstructions. This paper contributes to very few studies, which use the nested RCMs simulations for paleo climate studies and is suitable for publication in "Climate of the Past" . The authors should, however, consider my few points prior to final publication.

Main comments:

1- The usage of the stationary Kalman Gain shall be discussed in more details in the paper. It might be usable for time-slice simulations of several decades like this paper but for longer simulation windows which contain abrupt climatic shifts the static Kalman Gain can not capture the climate evolution. For centennial simulations, one shall use dynamic calculation of the background covariances. Such problems of static Kalman Gain must be discussed in the last section and the authors must mention what was the reason to go for static Kalman Gain.

2- I suggest to set up an extra test in which you explore the methodology, which you suggested in Page 11, lines 9-12. You might also split the 36 years of simulation and produce larger ensemble with more members, ie, transform any single simulation in a large background ensemble similar to the study of Hakim et al., 2016 . The reader is left with her/his curiosity to see if this might remove the trends in the RMSE. This might not be difficult, the DA is in offline mode and the results might be highly interesting for the community.

3- Instead of comparison of analysis with the gridded E-OBS data (Figures 11-12), I suggest to compare the analysis values with not assimilated observations. The gridded E-OBS already contains the assimilated observations and it makes the comparison very difficult.

4- The authors use the shifting of domain to create the ensemble members based on the reason that they do not touch the model configuration. However, different starting

times would also be a similar strategy. They can conduct a short test (with less than 10 years of simulation and with 2 or 3 members) to show if lag simulations also create comparable spread with the shifting of domains.

Minor comments:

1- Page 1 line 2: which kind of DA is expensive? Or calculation of covariance matrix in EnKF is expensive? Please revise.

2- Page 1 line 3: assimilation period or the time step of observations?

3- Page 1 line 14-16: too complicated. do you mean the radius within which we assume the observations are correlated?

4- Page 2 line 6-7 do you mean : "One of the main challenges is the lack of information for longer time-periods than the recent observed century" ? please modify!

5- Page 2 line 31: unclear to me, please explain uncertainty in what?

6- Page 3 lines 9 – 14. It should be mentioned that the offline DA is producing only the analysis state and the forecast skill which is the goal of usual DA (for example in weather predictions) is not considered. It is a kind of post-processing . From the text it seems the offline and online do the same job.

7- Page4 lines 1-5 : It should be made clear in the text that the state X is multivariable or just one variable and H is a linear function here.

8- Page 6 lines 13: "observationally constrained run"? I would call it "analysis". It is not a model run! It is manipulation of a model output.

9- Page 7 line 16: What do you mean by "optimal different boundaries"?

10- Page 7 line 21 the RMSE formula: change the superscripts Nature and Free and make it general. You also use this formula to calculate RMSE of analysis, etc. . .

11- Page 8 lines 20-29: could this minimum of RMSE be a local one? How is it set in

other studies?

12- Page 8 line 30: what do you mean "by the shape of the white noise added to create the pseudo-observation data" ? They are all White Noise. Clarify this!

13- Page 9 lines 28-30 As I mentioned in the main comments, the E-OBS uses kriging method for bringing observation on grids and comparison of analysis vs gridded E-OBS might not show the real skill. I suggest comparison of analysis against not assimilated observations.

14- Page 11 lines 16-20: Here a discussion on the culprits of stationary Kalman Gain is missing.

---

## Referee Comment (RC2) · Anonymous Referee #2 · 18 Dec 2017

In this paper, the authors test an offline DA approach using a high resolution regional climate model. The experiments test how error is reduced by assimilating pseudo and real observations.

The experiments are scientifically sound, but I have significant concerns about the applicability of the results in the current manuscript to the general paleoclimate reconstruction problem. What the authors have done in showing error reduction in some idealized reconstructions is a necessary first step in showing that the DA works, but I don't think the results shown here warrant publication. Many other previous studies have shown that DA for paleoclimate at a range of time and spatial scales works

(monthly to decadal and coarse to $\sim$1 degree resolution), so that's not really in question. In so far as the applicability is concerned, the experiments use a very dense observation network with very high SNR values, while the paleoclimate reconstruction problem has the opposite characteristics: low observation density and low SNR values. Typically reconstructions of this kind use a network based on actual proxy sites and SNR values of around 0.5. These choices have significant impacts on the skill of the reconstructions and the some of the conclusions that can be drawn from them. If a dense proxy network and a larger SNR are chosen, then this needs to be vigorously defended based on the scientific goals of the study.

However, with the current climate model simulations they have, the authors are well positioned to answer some important questions that would be directly relevant to paleoclimate reconstructions. Such questions include:

What benefits come from using the very high resolution simulations compared to the simulations that people have used so far? Can you get better reconstructions using very high resolution climate models? Are reconstructions that focus on specific regions more skillful than reconstructions designed on a global scale? Are certain variables better reconstructed in a regional framework? Etc.

So while I recommend that the paper not be published, I would strongly encourage the authors to resubmit the paper using reconstructions that are more clearly connected to the larger paleoclimate reconstruction problem.

Additional minor comments:

Section 3.2: Where does the localization function come from? What are x, y, and n? Does this function have compact support? Also, I don't think that one can choose an "optimal" localization independent of information about the observations.

The ensemble size appears to change between experiments. Is it possible to keep it the same size for all the experiments?

The figure captions are rather sparse. I'd recommend further explanation of the plots in the captions.

Many of the equations could benefit from a more condensed notation instead of writing out fully "Analysis" or "Trace", for example.

---

## Referee Comment (RC3) · Anonymous Referee #3 · 20 Dec 2017

The paper describes an approach for high-resolution climate reconstruction using an off-line assimilation of proxies into a set of regional climate model simulations. The set-up is tested with the COSMO-CLM model and a number of sensitivity studies are carried out. While the study is interesting, it somehow stops at a point that is still too distant from applications, and it is not clear what sort of applications the authors have in mind. I think the authors should better demonstrate how high-resolution climate reconstruction actually will be obtained and how they will be applied. Furthermore, I found the methodology not very well explained. However, because this is the first paper I am aware of that applies paleo DA to regional climate, I think we can learn a lot and therefore, in my view, the paper is potentially worth being published after major

revisions.

Major comments:

1) The design of the study is still far from a real world sparse proxy network. Just one example: They assimilate 500 observations, they even go up to 2700 stations and consider 100 "a small number of stations". I would be far more curious to see how the approach works with fewer observations, and what the author's view is concerning other variables (precipitation). Will this eventually work for tree rings? The results section is only 2 pages (part of which, i.e., the localisation, should actually be in the methods section).

2) The paper places itself in the sequence of recent work on paleo DA - it does not mention existing 0.5°-resolved statistical reconstructions. The motivation of many of the global paleo DA studies is to obtain a physically consistent global climate for time periods with spatially very heterogeneous coverage. There are good reasons for suspecting the same on the regional level, particularly for Europe (given the orography and land-sea contrast), but I think this needs to be better justified.

3) The methodology could be explained better. I already stumbled over p.4/l.5, which I first read as impliyng that X_TRUE and X_A is the same (is an "and" missing?). The terms X_NATURE and "free ensemble run" appear before they are introduced. There are some other instances (listed below).

Minor

P. 1, l.12: How can the selection of proxies reduce the background error?

p. 2, l.20: states

p. 3, l.13: Since the sentence cites DA approaches that were actually "applied", it might be good to cite Franke et al. (Scientific Data, 2017).

p. 3, l.20: The sentence is somehow odd: "optimum" in the first part implies a choice,

"truly" implies an esimation.

p. 5, eq. 8: X_NATURE is not introduced yet

p. 5, l. 5: X_Analysis was called X_A before

p. 5, l. 9: X_Analsis -> X_Analysis

p. 5, eq. 11 and 12 are both said to represent "the error covariance of the analysis"

p. 6, l. 9: Is the added noise spatially uncorrelated?

p. 7, l. 14: When describing the shift, the state vector should be defined (because it can no longer include the entire model domain - is it the "evaluation domain", which on my first reading I interpreted as the domain in which evaluations are done).

p. 7, l. 19: The analysis skill should be in the title, and mentioned in the text upfront. Some measure of dispersiveness might be interesting.

p. 7, l. 20: What is a "free ensemble run"? This term is not introduced.

P. 9, l. 13: Please explain the "universal behaviour of fluctuations of terrestrial near-surface temperature"

---

## Author Comment (AC1) · 27 Feb 2018

**Answer to anonymous referee #1's commnets**

Fallah et al.

February 27, 2018

We wish to thank the reviewer for his/her critics which definitely helped us a lot to make our point clearer. We will answer the comments(*italic*) point by point (**Bold**) in the following :

*1- The usage of the stationary Kalman Gain shall be discussed in more details in the paper. It might be usable for time-slice simulations of several decades like this paper but for longer simulation windows which contain abrupt climatic shifts the static Kalman Gain can not capture the climate evolution. For centennial simulations, one shall use dynamic calculation of the background covariances. Such problems of static Kalman Gain must be discussed in the last section and the authors must mention what was the reason to go for static Kalman Gain.*

**We agree with you and will add a discussion on stationary KG in the new manuscript . However, for time-slice simulations using RCMs which are conducted at high resolutions, it is very expensive to go over several decades. Our main focus in this study is contributed to time-slices of 30 years. This period length is chosen as representative time span for a typical climate. If there exists a regional RCM simulation longer than this period, the methodology can be applied more frequent than 30 years, eg. every decade or every 5 years.**

*2- I suggest to set up an extra test in which you explore the methodology,*

*which you suggested in Page 11, lines 9-12. You might also split the 36 years of simulation and produce larger ensemble with more members, ie, transform any single simulation in a large background ensemble similar to the study of Hakim et al., 2016 . The reader is left with her/his curiosity to see if this might remove the trends in the RMSE. This might not be difficult, the DA is in offline mode and the results might be highly interesting for the community.*

We implemented your comment. We took 40 random states from the climate pool of 4 members $\times$ 36 years of RCM simulations for each time-step and repeated the experiment instead of using the state of 4 members at the exact assimilation time. The results for winter(DJF) are shown in figure (1). Using random states as the background removed the uprising trend in the RMSE (red line in Fig.1.d) and the spread of the error is also reduced. However, the RMSE mean is in the range of the background state and there is no sign of error reduction in the analysis. This is more clear in the maps of analysis' RMSEs (Fig.1.b). Compared to the original background (Fig.1.a), using random states destroys the skill of the background itself (for example over west of the domain,i.e., Spain, Portugal, France, Morocco, ...) and reduction of RMSE elsewhere is leveled off. Therefore, usage of random states would be beneficial if the model had no significant skills at any region. However, the significant skill of the model background might be a characteristic of this particular RCM. We plan to add this experiment to the supplementary part of the new manuscript.

*3- Instead of comparison of analysis with the gridded E-OBS data (Figures 11-12), I suggest to compare the analysis values with not assimilated observations. The gridded E-OBS already contains the assimilated observations and it makes the comparison very difficult.*

It is true. According to reviewer 2 and 3's comments who were interested to see a real application of our methodology in paleoclimate, we decided to change this chapter. We will show results of assim-

(a) **Forecast of original ensemble back-ground**

(b) **Analysis using random background**

(c) **Original**

(d) **Using random background**

Figure 1: **(a)**: 36 years average of ensemble RMSE of the original simulations (without assimilation), **(b)**: 36 years averaged of RMSE of analysis using 40 random states as background, **(c)**: fieldmean of RMSE from ensemble (shading shows the ensemble spread, the white line the mean) and from analysis (black line), dashed lines are the linear fits. **(d)**: fieldmean of RMSE from ensemble (shading shows the ensemble spread, the white line the mean) and from analysis using random background states (red line), dashed lines are the linear fits and the blue line shows the 0.3 K RMSE value (is plotted only for comparison of the trends).

**ilation of pollen-based reconstruction within the RCM runs during the Holocene and remove the tests with E-OBS data. In the new experiments we assimilate 78% of the data and hold 22% for the validation. For more information please refer to our answer to question 5 of reviewer 2.**

*4- The authors use the shifting of domain to create the ensemble members*

*based on the reason that they do not touch the model configuration. However, different starting times would also be a similar strategy. They can conduct a short test (with less than 10 years of simulation and with 2 or 3 members) to show if lag simulations also create comparable spread with the shifting of domains.*

**We set up 4 new runs of 20 years each with 1 month lag starting time. Figure 2 shows the time average of ensemble spread for winter and summer. The ensemble spread patterns for both summer and winter are very similar to the domain-shifting experiments (Fig.2.c,d). However, the spread values are smaller, which might be due to the fact that the spread is larger in 36 years (domain-shifting experiments) than 20 years (time-lagged experiments).**

[Figure]

Figure 2: 20-year ensemble spread for winter(DJF) **(a)** and summer(JJA) **(b)** with lagging of initial time along with the 36-year ensemble spread for domain-shifting experiment winter(DJF) **(c)** and summer(JJA) **(d)**

*minor comments:*

*1- Page 1 line 2: which kind of DA is expensive? Or calculation of covariance matrix in EnKF is expensive? Please revise.*

**Done.**

*2- Page 1 line 3: assimilation period or the time step of observations?*

**We clarified in new version.**

*3- Page 1 line 14-16: too complicated. do you mean the radius within which we assume the observations are correlated?* **Done.** *5- Page 2 line 31: unclear to me, please explain uncertainty in what?* **We clarify this in the new version of the paper.**

---

## Author Comment (AC2) · 27 Feb 2018

**Answer to anonymous referee #2's commnets**

**Fallah et al.**

**February 27, 2018**

We would like to thank the reviewer for his comments and general feedbacks that gave us the opportunity to reconsider and eventually improve several aspects of the paper. In particular, following her/his comments, we realized that the goals of the paper were not particularly clear in the manuscript and that the reviewers were expecting a real paleoclimate application, for which the manuscript was initially designed. For this purpose, we will change the structure of the paper as follows:

1- introduction, 2- data and method, 3- Results: 3.1- Unconstrained Ensemble Runs, 3.2 Constrained Ensemble Runs (Perfect model experiment), 3.3 Application to a paleo study: the case of summer temperatures over Europe at the mid-Holocene, 4- discussions and conclusions

The case study we selected for the application of the proposed DA offline method, is one that has been the subject of a long-standing debate within the paleoclimate community, with contrasting interpretations arising from climate model simulations and from different proxy types. Indeed, it is particularly suitable for the scope of paleodata assimilation: joining together climate records and physical representation of the climate system could offer a more reliable picture than single proxy-datasets or climate models, and contribute to such

complex debates. We make the new intents of the paper clear and, in particular, that our focus is to introduce, test and apply a data assimilation method to a specific paleoclimate case-study. Our answer contains 11 new figures located at the end of this answer. We answer the comments(*italic*) point by point in Bold:

*1- In this paper, the authors test an offline DA approach using a high resolution regional climate model. The experiments test how error is reduced by assimilating pseudo and real observations. The experiments are scientifically sound, but I have significant concerns about the applicability of the results in the current manuscript to the general paleoclimate reconstruction problem. What the authors have done in showing error reduction in some idealized reconstructions is a necessary first step in showing that the DA works, but I don't think the results shown here warrant publication. Many other previous studies have shown that DA for paleoclimate at a range of time and spatial scales works (monthly to decadal and coarse to ∼ 1 degree resolution), so that's not really in question. In so far as the applicability is concerned, the experiments use a very dense observation network with very high SNR values, while the paleoclimate reconstruction problem has the opposite characteristics: low observation density and low SNR values. Typically reconstructions of this kind use a network based on actual proxy sites and SNR values of around 0.5. These choices have significant impacts on the skill of the reconstructions and the some of the conclusions that can be drawn from them. If a dense proxy network and a larger SNR are chosen, then this needs to be vigorously defended based on the scientific goals of the study.*

The aim of our work is not the repetition of experiments done in previous studies. We propose a method for data assimilation that could be applied to paleoclimate studies. In particular we consider the case of mid Holocene for Europe since it is one of the most important case of debate between climate modelers and proxy community. We select the data of [Russo and Cubasch, 2016] since they are one of the most recent simulations for the period. This choice does not exclude the possibility to apply the Offline DA method to other periods or simulations. This case study is done with realistic proxy data (numbers, locations, SNR, ...), therefore we believe that we cover all your critics mentioned above.

In the paper we have already done series of DA experiments for winter and summer with different SNR values starting from 0.1 to 10. Figure 4 of the submitted manuscript shows the results with different SNR values. You mentioned that the SNR value of 3 is very high in our experiment set-up. However, the difference in field-mean RMSE for SNR=0.1 and SNR=3.0 in summer is not significant (<0.003 K) and for Winter (<0.01 K). For clarity we have plotted again the RMSE map for SNR=0.1 and SNR=3.0 for winter and summer here (Fig.1 and Fig.2). The RMSE maps show very little differences between SNR=0.5 and SNR=3.0. We hoped that the figure 4 was clear for the readers, if that is not the case, we could add the maps along with the figure 4 in new version of the manuscript.

Regarding the number of assimilated observations, we had uploaded two movies in the assets part of the publication, which show changes in RMSE with increasing number of observations from 100 to 2700. We repeated the DA experiment 54 times, 2(seasons)× 27(sets of stations). It was already shown in the videos that with even small number of observations the analysis has still significant skill. Here we plot the RMSEs again for 100 to 400 assimilated observations for winter and summer (Figs 3 and 4). On the other hand in very recent studies focusing on reconstruction of climatic variables the number of records used are similar to the number we have chosen: for example [Mauri et al., 2015] have used "879 selected pollen sites representing nearly 60,000 pollen counts" (see Figure 1 of their paper `https://ars.els-cdn.com/content/image/1-s2.0-S0277379115000372-gr1.jpg`). Or [Franke et al., 2017] have used a proxy network which is very dense over Europe (Figure 3 of their paper :`https://www.nature.com/articles/sdata201776/figures/3`). Or the location of sites in the study of [Marlon et al., 2017] in North

America is also a dense one. Or [Cook et al., 2010b] where they used 1,854 annual tree-ring chronologies over North America (**Figure 5 of their study** : `http://onlinelibrary.wiley.com/doi/10.1002/jqs.1303/pdf`). Or the study of [Cook et al., 2010a] where they used a **327-series** tree-ring chronology network to reconstruct the Palmer Drought index over Asian mansoon area (**Figure 1 of their paper** : `https://d2ufo47lrtsv5s.cloudfront.net/content/sci/328/5977/486/F1.large.jpg`). Therefore, we believe **500 stations over Europe looks realistic with new advances in paleo-data collection, synthesis and stewardship** (for example `https://www.ncdc.noaa.gov/data-access/paleoclimatology-data/datasets`). **Even if that is not the case, we showed fewer numbers of observations still work for error reduction of analysis.** Please note that we do not only show the results of perfect model experiments, but on the 3rd set-up we are assimilating the real temperature data from E-OBS.

However, to clear any further doubts, we set up a new experiment in which we assimilated real observation during the Holocene. Please read the answer to question 5.

*2- However, with the current climate model simulations they have, the authors are well positioned to answer some important questions that would be directly relevant to paleoclimate reconstructions. Such questions include: What benefits come from using the very high resolution simulations compared to the simulations that people have used so far?*

This question is challenging the usage of nested RCM simulations for paleo-climate studies or if RCM simulations have additional value compared to the GCM simulations. Generally speaking, such questions might be a topic for a new study and it is out of the scope of our research in this paper. However, the questions regarding the advantage of RCMs over GCMs for paleoclimate, relatively to the considered case study (temperature over europe at the mid Holocene) has been answered in [Russo and Cubasch, 2016]. Even though the

paper did not solve the issue, it gives a relatively important support to the choice of the CCLM simulations for the purposes of our new study. There exist several methods for testing the added values of RCM simulations in capturing more details of the climate state. In the context of offline DA, the background should be a dynamically consistent state of the climate. It might come from a random draw from any climatology or from ensemble of free simulations (without re-initializations). There is no limit on the background to be global or regional. In most of the Ensemble Kalman Filtering approaches one considers covariance localization to remove spurious correlations and the covariance is not calculated globally. If the background can capture more details of the climate correctly, it might add value to the analysis as well. Here the offline DA plays as a postprocessing agent and tries to correct the background and if RCM background is more skilful than in the GCM simulations, is out of our research scope. However, we present some discussions in the answer to the next two questions.

*3- Can you get better reconstructions using very high resolution climate models?*

This question is very close to the previous one. It is a question about which benefits one can get by using RCM simulations. The resolution of the reconstructed field is the one of the background state. Usually the usage of RCM simulations in pealo studies are bounded in the time integragtions of several decades which is called time-slice simulation methods. Where one needs to explore the climate state over a specific region with more details. Studies using the GCM simulations as their background, however, are mostly interested in longer time-scales of several thousand years, and the climatic interconnections. The approach presented here is an additional tool for magnifying the interesting time-slices within the driving GCM with more details which are not captured by GCMs locally. In practice it would be very beneficial to assimilate the observations in both the

RCM and the driving GCM.

*4- Are reconstructions that focus on specific regions more skillful than reconstructions designed on a global scale? Are certain variables better reconstructed in a regional framework? Etc.*

This could be tested in an additional study. However, one should first set the skill metric for such comparisons. Most of the paleo GCM simulations (if fully coupled) are at low/middle resolutions and number of the ensemble members are very limited. If the RCM can capture more realistic details of the climate state, then it would be beneficial to use them for time-slice simulations. We should mention that one of our main motivations is to resolve the gap of regional to local scale climate change, which might be of interest in the paleo-community. For example the uncertainties of proxy data are bound at regional scales. However, the performance of the RCM might vary for different variables. Assuming that in an RCM there are more realistic physical processes implemented than the GCM (especially complex topography) which otherwise had to be parameterised in GCMs, the resolution of such models are of advantage when comparing with the proxy data. One problem of using coarse GCM data might be for example the process of selecting the best observation within a grid cell for DA scheme ("data thinning") or sampling with averaging for the observations within a grid cell ("super-obing"). By using the RCMs, we are reducing such problems. For comparison of proxy and GCM, one might use classical approaches (statistical downscaling, upscaling, forward model), however, such methods for a coarse resolution of GCM might be very challenging as one has to evaluate or train such models with very short observation time-window. The proxy-data relation might also change over time (not stationary). On the other hand, the feedback of regional climatic changes on the global scale is ignored using one-way nesting approach applied in our simulations. Finally, we should mention that with our domain set up the RCM is constrained by the GCM at the lateral boundaries and therefore its

internal variability is similar to the driving GCM at large scale. This behavior is detectable by the maps of ensemble spread shown in the manuscript and answer to reviewer 1. By changing the domain or initialization time, the RCM simulations do not vary dramatically.

*5- So while I recommend that the paper not be published, I would strongly encourage the authors to resubmit the paper using reconstructions that are more clearly connected to the larger paleoclimate reconstruction problem.*

We would like to ask you to give us a second chance to modify the manuscript accordingly. We hope the revised version could be accepted in the CP. As recommended, we have done additional experiments using real proxies(pollen-based) and precomputed COSMO-CLM simulations during several time-slices of the Holocene. A complete explanation of the experiments will be implemented in the new version of the manuscript. Here we briefly explain the method and show the results for one summer-time (JJA) time-slice of 6000 years before present(6KBP):

We have used the pollen-based temperature reconstructions of [Mauri et al., 2015], (for more information on the data ref. to [Mauri et al., 2015] and the references therein) and the model simulations of [Russo and Cubasch, 2016]. For evaluation, we keep 22% of the proxy data as the test data, and assimilate the rest 78%. The test data is selected randomly. We tested two approaches for averaging the proxy data for the target time-slices: a) averaging with respect to their distance to the target year (6KBP) and b) averaging with respect to their uncertainties provided as standard error by [Mauri et al., 2015] at the observation time:

*a):* We chose a time window centered on the target year, (e.g., reference time $\mp 500$ years) and weight the values and the standard errors by their time distance to the target year. The weights are chosen from a normal distribution with standard deviation of 100. Totally 3 weighting time-spans are defined. Figure 5 shows the weights

assigned to each time interval with respect to the reference time.

*b):* We chose the observations within the time window of reference year $\mp 500$ years. Then we apply the weighted arithmetic mean of the temperature and its standard error to calculate the time-slice values (6KBP). Each proxy is weighted first by its standard error. Then the weighted mean is calculated by:

$$\bar{x} = \frac{\sum_{i=1}^{n}(x_i \sigma_i^{-2})}{\sum_{i=1}^{n} \sigma_i^{-2}} \tag{1}$$

The uncertainty of the weighted mean is given then by:

$$\sigma_{\bar{x}}^2 = \frac{1}{\sum_{i=1}^{n} \sigma_i^{-2}} \tag{2}$$

Where the $\sigma$ is the standard error. **Figure 6** shows the schematic of weighting of observations with respect to their standard errors.

Finally, for the model, we assign the **25-year time-average** as the expected value and the standard deviation from the mean as uncertainty measure. **Figure 7** shows the schematic of the approach. Please note that for each time-slice we have a single model run of **25 years**. We assume that each year of the model simulation could serve as an ensemble member for each target time-slice (6KBP). Therefore, the analysis is done for a single step using the background information of the **25 years**.

**DA Results:**

Figure 8.a shows the analysis results for summer T2m temperature anomalies (6KBP-0.2KBP) over Europe along with the testing observations (circles) superposed on their standard error(squares). The analysis and the proxy show a good agreement especially for observations with low standard error. In contrast, the model forecast (without assimilation) and the proxies (Fig. 8.b) show little agreement. The assimilated observation are shown in Figure 9. The positive anomalous region over Romanian in the analysis is due to the cluster of proxy data with low standard error over this region.

Overall, we conclude that the proposed DA approach is contributing to the error reduction in the analysis values where the pure model outputs might not capture the local patterns.

Repeating the analysis approach with the different weighted mean of observations (method b) leads to very similar patterns in analysis (Figures 10 and 11). In the final version of the manuscript we will publish the maps and the netcdf files of the analysis and model for summer and winter for 6 different time slices during the Holocene (6KBP to 1KBP).

*Additional minor comments: Section 3.2: Where does the localization function come from? What are x, y, and n? Does this function have compact support? Also, I don't think that one can choose an "optimal" localization independent of information about the observations.*

We will move this part to the OI basics where the P, x, y and are defined and will work on this comment in the new version of the manuscript.

*The ensemble size appears to change between experiments. Is it possible to keep it the same size for all the experiments?*

We could remove the experiment with short integration runs, however we thought that the effect of ensemble size might be interesting for the readers. On the other hand reviewer 1 is asking to set up a new sets of experiment with increasing the ensemble size for longer simulations by using random draws from climatology.

*The figure captions are rather sparse. I'd recommend further explanation of the plots in the captions. Many of the equations could benefit from a more condensed notation instead of writing out fully "Analysis" or "Trace", for example.*

Done in new version of manuscript.

**References**

[Cook et al., 2010a] Cook, E. R., Anchukaitis, K. J., Buckley, B. M., D'Arrigo, R. D., Jacoby, G. C., and Wright, W. E. (2010a). Asian monsoon failure and megadrought during the last millennium. *Science*, 328(5977):486–.

[Cook et al., 2010b] Cook, E. R., Seager, R., Heim, R. R., Vose, R. S., Herweijer, C., and Woodhouse, C. (2010b). Megadroughts in north america: placing ipcc projections of hydroclimatic change in a long-term palaeoclimate context. *J. Quaternary Sci.*, 25(1):48–61.

[Franke et al., 2017] Franke, J., Brönnimann, S., Bhend, J., and Brugnara, Y. (2017). A monthly global paleo-reanalysis of the atmosphere from 1600 to 2005 for studying past climatic variations. *Scientific Data*, 4:170076–.

[Marlon et al., 2017] Marlon, J. R., Pederson, N., Nolan, C., Goring, S., Shuman, B., Robertson, A., Booth, R., Bartlein, P. J., Berke, M. A., Clifford, M., Cook, E., Dieffenbacher-Krall, A., Dietze, M. C., Hessl, A., Hubeny, J. B., Jackson, S. T., Marsicek, J., McLachlan, J., Mock, C. J., Moore, D. J. P., Nichols, J., Peteet, D., Schaefer, K., Trouet, V., Umbanhowar, C., Williams, J. W., and Yu, Z. (2017). Climatic history of the northeastern united states during the past 3000 years. *Clim. Past*, 13(10):1355–1379.

[Mauri et al., 2015] Mauri, A., Davis, B., Collins, P., and Kaplan, J. (2015). The climate of europe during the holocene: a gridded pollen-based reconstruction and its multi-proxy evaluation. *Quaternary Science Reviews*, 112(Supplement C):109–127.

[Russo and Cubasch, 2016] Russo, E. and Cubasch, U. (2016). Mid-to-late holocene temperature evolution and atmospheric dynamics over europe in regional model simulations. *Clim. Past*, 12(8):1645–1662.

(a) **SNR=0.5**

[Figure]

(b) **SNR=3.0**

[Figure]

Figure 1: **(a)**: 10-year RMSE [K] of analysis for summer(JJA) with SNR=0.5 and **(b)** SNR=3.0.

(a) **SNR=0.5**

[Figure]

(b) **SNR=3.0**

[Figure]

Figure 2: **(a)**: 10-year RMSE [K] of analysis for winter(DJF) with SNR=0.5 and **(b)** SNR=3.0.

(a) **Without Assimilation for summer**

[Figure]

(b) **Number of Obs. = 100**     (c) **Number of Obs. = 200**

(d) **Number of Obs. = 300**     (e) **Number of Obs. = 400**

[Figure]

Figure 3: Analyis RMSE for different number of observations for summer (JJA)
**(a)**: without assimilation, **(b)**; with 100, **(c)**: with 200, **(d)**: with 300 and **(e)**: with 400 observations.

(a) **Without Assimilation for winter**

[Figure]

(b) **Number of Obs. = 100**           (c) **Number of Obs. = 200**

(d) **Number of Obs. = 300**           (e) **Number of Obs. = 400**

[Figure]

Figure 4: Analyis RMSE for different number of observations for winter (DJF) **(a)**: without assimilation, **(b)**; with 100, **(c)**: with 200, **(d)**: with 300 and **(e)**: with 400 observations.

[Figure]

Figure 5: Schematic showing the weights for observations with respect to their distance to the target year. Time window of 1000 years is chosen. The observations are weighted depending on their distances to the reference time.

[Figure]

Figure 6: Schematic showing the weights for observations with respect to their standard error. Time window of 1000 years is chosen. The red dots resemble the proxies in the 1000-year time window and the green dot resembles the weighted mean.

[Figure]

Figure 7: Schematic showing how the expected value (the mean) and the deviation from the mean for each time-slice simulation is selected. The model simulation is 25 years long. Green line resembles the model state.

(a) **analysis**

[Figure]

(b) **model**

[Figure]

Figure 8: DA results using weighted arithmetic mean by time distances: **(a)**: Analysis values along with the testing observations (circles) and their standard error (squares). Values with error covariance of analysis greater than 0.9 are masked out from analysis, and **(b)** the model forecast.

[Figure]

Figure 9: Weighted arithmetic mean using time distances: Anomalies ((6KBP - 0.2KBP)) of assimilated observations (circles) superposed on their standard errors(squares) with values in K. Color-bar of the standard errors as in Figure 8.a.

(a) **analysis**

[Figure]

(b) **model**

[Figure]

Figure 10: DA results using weighted arithmetic mean by standard errors: **(a)**: Analysis values along with the testing observations (circles) and their standard error (squares). Values with error covariance of analysis greater than 0.9 are masked out from analysis, and **(b)** the model forecast.

[Figure]

Figure 11: Weighted arithmetic mean using standard errors: Anomalies ((6KBP - 0.2KBP)) of assimilated observations (circles) superposed on their standard errors(squares) with values in K. Color-bar of the standard errors as in Figure 8.a.

---

## Author Comment (AC3) · 27 Feb 2018

**Answer to anonymous referee #3's commnets**

Fallah et al.

February 27, 2018

We wish to thank the reviewer for his/her critics and positive feedback. We will answer the comments(*italic*) point by point in the following (**Bold**) :

*1- The paper describes an approach for high-resolution climate reconstruction using an off-line assimilation of proxies into a set of regional climate model simulations. The set-up is tested with the COSMO-CLM model and a number of sensitivity studies are carried out. While the study is interesting, it somehow stops at a point that is still too distant from applications, and it is not clear what sort of applications the authors have in mind. I think the authors should better demonstrate how high-resolution climate reconstruction actually will be obtained and how they will be applied. Furthermore, I found the methodology not very well explained. However, because this is the first paper I am aware of that applies paleo DA to regional climate, I think we can learn a lot and therefore, in my view, the paper is potentially worth being published after major revisions.*

**We wish to thank you for your suggestions. According to your and reviewer 2's concerns on the real application of the methodology and how such high resolution information will be obtained, we set up new experiments using pre-computed time-slice COSMO-CLM simulations over Europe during the Holocene and pollen-based temperature reconstructions. Please refer to our answer to reviewer 2's 5th question. We hope that the new manuscript is closer to real application and will be accepted for climate of the past. The appli-**

cation of such high resolution climate data for the paleo-modeling community, might be of interest for example for evaluation of coupled simulations against high resolution climate maps over the target region (here Europe).

*1) The design of the study is still far from a real world sparse proxy network. Just one example: They assimilate 500 observations, they even go up to 2700 stations and consider 100 "a small number of stations". I would be far more curious to see how the approach works with fewer observations, and what the author's view is concerning other variables (precipitation). Will this eventually work for tree rings? The results section is only 2 pages (part of which, i.e., the localisation, should actually be in the methods section).*

This comment is similar to reviewer 2's comment number 1. We briefly answer it again here: In very recent studies focusing on reconstruction of climatic variables the number of records used are similar to the number we have chosen: for example [Mauri et al., 2015] have used "879 selected pollen sites representing nearly 60,000 pollen counts" (see Figure 1 of their paper `https://ars.els-cdn.com/content/image/1-s2.0-S0277379115000372-gr1.jpg`). Or [Franke et al., 2017] have used a proxy network which is very dense over Europe (Figure 3 of their paper :`https://www.nature.com/articles/sdata201776/figures/3`). Or the location of sites in the study of [Marlon et al., 2017] in North America is also a dense one. Or [Cook et al., 2010b] where they used 1,854 annual tree-ring chronologies over North America (Figure 5 of their study : `http://onlinelibrary.wiley.com/doi/10.1002/jqs.1303/pdf`). Or the study of [Cook et al., 2010a] where they used a 327-series tree-ring chronology network to reconstruct the Palmer Drought index over Asian mansoon area (Figure 1 of their paper : `https://d2ufo47lrtsv5s.cloudfront.net/content/sci/328/5977/486/F1.large.jpg`). Therefore, we believe 500 stations over Europe looks realistic with new advances in paleo-data collection, synthesis and stewardship (for example `https://www.ncdc.noaa.gov/data-access/paleoclimatology-data/datasets`). Following your first comment we also assimilated a proxy

number of $\sim 300$ in the real application.

On your comment about precipitation or usage of tree rings, we should add that here we use an inverse model's outcome (temperature reconstructions) and one can also use a proxy forward model of any kind and repeat the assimilation on proxy space instead of the model space. We will add more detail on that in the discussion. We have used such an approach in a recent study ([Acevedo et al., 2017]) with an extended version of Vaganov–Shashkin Lite (VSL) process-based tree-ring-width forward model ([Tolwinski-Ward et al., 2011]) and the SPEEDY climate model. The error reduction for the precipitation was shown to be not significant using the Enkf. However, the model was of intermediate complexity in that study.

*2) The paper places itself in the sequence of recent work on paleo DA - it does not mention existing 0.5 -resolved statistical reconstructions. The motivation of many of the global paleo DA studies is to obtain a physically consistent global climate for time periods with spatially very heterogeneous coverage. There are good reasons for suspecting the same on the regional level, particularly for Europe (given the orography and land-sea contrast), but I think this needs to be better justified.*

We hope that by showing the real application with RCM simulations throughout the Holocene and adding the references of the pollen data, especially the work of [Mauri et al., 2015], we will cover this comment. We will add a paragraph of previous statistical reconstruction efforts on Europe in the introduction of the new manuscript. On the comment on the usage of RCM instead of GCM in paleo-data assimilation, we should mention that one of our main motivations is to resolve the gap of regional to local scale climate change, which might be of interest in the paleo-community. For example the uncertainties of proxy data are bound at regional scales. However, as you mentioned in question 1, the performance of the RCM might vary for different variables. Assuming that in an RCM there are more realistic

physical processes implemented than the GCM (especially complex topography) which otherwise had to be parameterised in GCMs, the resolution of such models are of advantage when comparing with the proxy data. One problem of using coarse GCM data might be for example the process of selecting the best observation within a grid cell for DA scheme ("data thinning") or sampling with averaging for the observations within a grid cell ("super-obing"). By using the RCMs, we are reducing such problems. For comparison of proxy and GCM, one might use classical approaches (statistical downscaling, upscaling, forward model), however, such methods for a coarse resolution of GCM might be very challenging as one has to evaluate or train such models with very short observation time-window. The proxy-data relation might also change over time (not stationary). On the other hand, the feedback of regional climatic changes on the global scale is ignored using one-way nesting approach applied in our simulations. Finally, we should mention that with our domain set up the RCM is constrained by the GCM at the lateral boundaries and therefore its internal variability is similar to the driving GCM at large scale. This behavior is detectable by the maps of ensemble spread shown in the manuscript and answer to reviewer 1. By changing the domain or initialization time, the RCM simulations do not vary dramatically for the averaged seasonal maps.

*3) The methodology could be explained better. I already stumbled over p.4/l.5, which I first read as impliyng that X_TRUE and X_A is the same (is an "and" missing?). The terms X_NATURE and "free ensemble run" appear before they are introduced. There are some other instances (listed below).*

Thanks. We will go through the formulation of mathematical terms and describe them as they appear in the text. We take care of all your minor comments in the new version of the manuscript.

**1 Minor comments**

*P. 1, l.12: How can the selection of proxies reduce the background error?* **We change the text accordingly. The analysis error is reduced compared to the background.**

*p. 2, l.20: states* **Changed.**

*p. 3, l.13: Since the sentence cites DA approaches that were actually "applied", it might be good to cite Franke et al. (Scientific Data, 2017).* **Done.**

*p. 3, l.20: The sentence is somehow odd: "optimum" in the first part implies a choice, C2"truly" implies an esimation.* **Changed.**

*p. 5, eq. 8: X_NATURE is not introduced yet p. 5, l. 5: X_Analysis was called X_A before p. 5, l. 9: X_Analsis -¿ X_Analysis* **Done.**

*p. 5, eq. 11 and 12 are both said to represent "the error covariance of the analysis"* **Changed. The trace of $P^a$ is the total error variance of the analysis and $P^a$ is the error covariance of the analysis.**

*p. 6, l. 9: Is the added noise spatially uncorrelated?* **They could be correlated. It is common practice to assumed that they are uncorrelated for the sake of simplicity and affordability.**

*p. 7, l. 14: When describing the shift, the state vector should be defined (because it can no longer include the entire model domain - is it the "evaluation domain", which on my first reading I interpreted as the domain in which evaluations are done).* **We will describe it clearer. It was previously described in the caption of the figure 1.**

*p. 7, l. 19: The analysis skill should be in the title, and mentioned in the text upfront. Some measure of dispersiveness might be interesting.* **Done.**

*p. 7, l. 20: What is a "free ensemble run"? This term is not introduced.* **A run without data assimilation. We will describe it clearer.**

*P. 9, l. 13: Please explain the "universal behaviour of fluctuations of terres-*

*trial near-surface temperature"* **Done. There exist a correlation between the temperature and the topography. The power-law behavior seen in topography also exists for the near surface temperatures. There exist a universal persistent role in the static geometry of the Earth which controls the dynamics of atmosphere.**

**References**

[Acevedo et al., 2017] Acevedo, W., Fallah, B., Reich, S., and Cubasch, U. (2017). Assimilation of pseudo−tree−ring−width observations into an atmospheric general circulation model. *Clim. Past*, 13(5):545–557.

[Cook et al., 2010a] Cook, E. R., Anchukaitis, K. J., Buckley, B. M., D'Arrigo, R. D., Jacoby, G. C., and Wright, W. E. (2010a). Asian monsoon failure and megadrought during the last millennium. *Science*, 328(5977):486–.

[Cook et al., 2010b] Cook, E. R., Seager, R., Heim, R. R., Vose, R. S., Herweijer, C., and Woodhouse, C. (2010b). Megadroughts in north america: placing ipcc projections of hydroclimatic change in a long-term palaeoclimate context. *J. Quaternary Sci.*, 25(1):48–61.

[Franke et al., 2017] Franke, J., Brönnimann, S., Bhend, J., and Brugnara, Y. (2017). A monthly global paleo-reanalysis of the atmosphere from 1600 to 2005 for studying past climatic variations. *Scientific Data*, 4:170076–.

[Marlon et al., 2017] Marlon, J. R., Pederson, N., Nolan, C., Goring, S., Shuman, B., Robertson, A., Booth, R., Bartlein, P. J., Berke, M. A., Clifford, M., Cook, E., Dieffenbacher-Krall, A., Dietze, M. C., Hessl, A., Hubeny, J. B., Jackson, S. T., Marsicek, J., McLachlan, J., Mock, C. J., Moore, D. J. P., Nichols, J., Peteet, D., Schaefer, K., Trouet, V., Umbanhowar, C., Williams, J. W., and Yu, Z. (2017). Climatic history of the northeastern united states during the past 3000 years. *Clim. Past*, 13(10):1355–1379.

[Mauri et al., 2015] Mauri, A., Davis, B., Collins, P., and Kaplan, J. (2015). The climate of europe during the holocene: a gridded pollen-based reconstruction and its multi-proxy evaluation. *Quaternary Science Reviews*, 112(Supplement C):109–127.

[Tolwinski-Ward et al., 2011] Tolwinski-Ward, S. E., Evans, M. N., Hughes, M., and Anchukaitis, K. J. (2011). An efficient forward model of the climate controls on interannual variation in tree-ring width. *Climate Dynamics*, 36(11-12):2419–2439.